# Policy Expansion for Bridging Offline-to-Online Reinforcement Learning

**Haichao Zhang**   **Wei Xu**   **Haonan Yu**
Horizon Robotics, Cupertino CA 95014
`{haichao.zhang, wei.xu, haonan.yu}@horizon.ai`

## Abstract

Pre-training with offline data and online fine-tuning using reinforcement learning is a promising strategy for learning control policies by leveraging the best of both worlds in terms of sample efficiency and performance. One natural approach is to initialize the policy for online learning with the one trained offline. In this work, we introduce a policy expansion scheme for this task. After learning the offline policy, we use it as one candidate policy in a policy set. We then expand the policy set with another policy which will be responsible for further learning. The two policies will be composed in an adaptive manner for interacting with the environment. With this approach, the policy previously learned offline is fully retained during online learning, thus mitigating the potential issues such as destroying the useful behaviors of the offline policy in the initial stage of online learning while allowing the offline policy participate in the exploration naturally in an adaptive manner. Moreover, new useful behaviors can potentially be captured by the newly added policy through learning. Experiments are conducted on a number of tasks and the results demonstrate the effectiveness of the proposed approach. Code is available: `https://github.com/Haichao-Zhang/PEX`.

## 1 Introduction

Reinforcement learning (RL) has shown great potential in various fields, reaching or even surpassing human-level performances on many tasks (e.g. Mnih et al., 2015; Silver et al., 2017; Schrittwieser et al., 2019; Tsividis et al., 2021). However, since the policy is learned from scratch for a given task in the standard setting, the number of samples required by RL for successfully solving a task is usually large, which limits its applicability in many practical scenarios such as robotics, where physical interaction and data collection has a non-trivial cost.

In many cases, there is a good amount of offline data that has already been available (Kober et al., 2013; Rastgoftar et al., 2018; Cabi et al., 2019), e.g., collected during previous iterations of experiments or from human (e.g. for the task of driving). Instead of ab initio learning as in the common RL setting, how to effectively leverage the already available offline data for helping with online policy learning is an interesting and open problem (Vecerík et al., 2017; Hester et al., 2018; Nair et al., 2018).

Offline RL is an active recent direction that aims to learn a policy by purely using the offline data, without any further online interactions (Fujimoto et al., 2019; Kumar et al., 2020; Fujimoto & Gu, 2021; Levine et al., 2020; Ghosh et al., 2022; Chen et al., 2021; Janner et al., 2021; Yang et al., 2021; Lu et al., 2022; Zheng et al., 2022). It holds the promise of learning from suboptimal data and improving over the behavior policy that generates the dataset (Kumar et al., 2022), but its performance could still be limited because of its full reliance on the provided offline data.

To benefit from further online learning, one possible way is to pre-train with offline RL, and warm start the policy of an online RL algorithm to help with learning and exploration when learning online. While this pre-training + fine-tuning paradigm is natural and intuitive, and has received great success in many fields like computer vision (Ge & Yu, 2017; Kornblith et al., 2019) and natural language processing (Devlin et al., 2018; Radford & Narasimhan, 2018; Brown et al., 2020), it is less widely used in RL. Many early attempts in RL community report a number of negative results along this direction. For example, it has been observed that initializing the policy with offline pre-training and then fine-tuning the policy with standard online RL algorithms (e.g. SAC (Haarnoja et al., 2018))

sometimes suffers from non-recoverable performance drop under certain settings (Nair et al., 2020; Uchendu et al., 2022), potentially due to the distribution shift between offline and online stages and the change of learning dynamics because of the algorithmic switch.

Another possible way is to use the same offline RL algorithm for online learning. However, it has been observed that standard offline RL methods generally are not effective in fine-tuning with online data, due to reasons such as conservativeness of the method (Nair et al., 2020). Some recent works in offline RL also start to focus on the offline-pre-training + online fine-tuning paradigm (Nair et al., 2020; Kostrikov et al., 2022). For this purpose, they share the common philosophy of designing an RL algorithm that is suitable for both offline and online phases. Because of the unified algorithm across phases, the network parameters (including those for both critics and actor) trained in the offline phase can be reused for further learning in the online phase.

Our work shares the same objective of designing effective offline-to-online training schemes. However, we take a different perspective by focusing on how to bridge offline-online learning, and not on developing yet another offline or online RL method, which is orthogonal to the focus of this work. We will illustrate the idea concretely by instantiating our proposed scheme by applying it on existing RL algorithms (Kostrikov et al., 2022; Haarnoja et al., 2018). The contributions of this work are:

- we highlight the value of *properly connecting* existing offline and online RL methods in order to enjoy the best of both worlds, a perspective that is alternative and orthogonal to developing completely new RL algorithms;
- we propose a simple scheme termed as *policy expansion* for bridging offline and online reinforcement learning. The proposed approach is not only able to preserve the behavior learned in the offline stage, but can also leverage it adaptively during online exploration and along the process of learning;
- we verify the effectiveness of the proposed approach by conducting extensive experiments on various tasks and settings, with comparison to a number of baseline methods.

## 2 PRELIMINARIES

We briefly review some related basics in this section, first on model-free RL for online policy learning, and then on policy learning from offline dataset.

### 2.1 ONLINE REINFORCEMENT LEARNING

Standard model-free RL methods learn a policy that maps the current state $s$ to a distribution of action $a$ as $\pi(s)$. The policy is typically modeled with a neural network $\pi_\theta(s)$ with $\theta$ denoting the learnable parameters. To train this policy, there are different approaches including on-policy (Sutton et al., 2000; Schulman et al., 2017) and off-policy RL methods (Lillicrap et al., 2016; Haarnoja et al., 2018; Fujimoto et al., 2018; Zhang et al., 2022). In this work, we mainly focus on off-policy RL for online learning because of its higher sample efficiency. Standard off-policy RL methods rely on the state-action value function $Q(s, a)$ using TD-learning:

$$Q(s, a) = r(s, a) + \gamma \mathbb{E}_{s' \sim T(s,a), a' \sim \pi_\theta(s')} \big[ Q(s', a') \big],$$

where $T(s, a)$ denotes the dynamics function and $r(s, a)$ the reward. $\gamma \in (0, 1)$ is a discount factor. By definition, $Q(s, a)$ represents the accumulated discounted future reward starting from $s$, taking action $a$, and then following policy $\pi_\theta$ thereafter.

The optimization of $\theta$ is achieved by maximizing the following function:

$$\max_\theta \mathbb{E}_{s \sim \mathcal{D}} \mathbb{E}_{a \sim \pi_\theta} Q(s, a). \tag{1}$$

where $\mathcal{D}$ denotes replay buffer for storing online trajectories. In the typical RL setting, learning is conducted from scratch by initializing all parameters randomly and interacting with the world with the randomly policy. $Q(s, a)$ can be implemented as a neural network $Q_\phi(s, a)$ with parameter $\phi$.

### 2.2 POLICY LEARNING FROM OFFLINE DATASET

Policy learning from offline datasets has been investigated from different perspectives. Given expert-level demonstration data, behavior cloning (BC) (Pomerleau, 1988; Bain & Sammut, 1996) is an

effective approach for offline policy learning because of its simplicity and effectiveness. In fact, in some recent work, BC has shown to perform competitively with some offline RL methods (Fujimoto & Gu, 2021; Chen et al., 2021). Given a dataset $\mathcal{D}_{\text{offline}} = \{(s_i, a_i)\}$ consisting of expert's state action pairs $(s_i, a_i)$, BC trains the policy with maximum likelihood over the data:

$$\max_\theta \mathbb{E}_{(s,a) \sim \mathcal{D}_{\text{offline}}} \log \pi_\theta(a|s). \tag{2}$$

Although BC has the benefits of reducing the policy learning task to an ordinary supervised learning task, it suffers from the well-known distributional shift issue (Codevilla et al., 2019; Muller et al., 2005; de Haan et al., 2019; Wen et al., 2020). Another limitation is that BC has a relatively strong requirements on the data quality, and is not good at learning from suboptimal data.

Offline RL is a category of methods that are more suitable for policy learning from noisy and suboptimal offline data (Kumar et al., 2022). When focusing on offline learning only, the core challenge is how to address the extrapolation error due to querying the critic function with out-of-distribution actions (Fujimoto et al., 2019; Kumar et al., 2020). Common strategies include constraining the actions to be close to dataset actions (Fujimoto et al., 2019; Fujimoto & Gu, 2021), and constraining the critic to be conservative for out of data distribution actions (Kumar et al., 2020). The recent implicit Q-learning (IQL) method (Kostrikov et al., 2022) addresses this issue by learning a value network to match the expectile of the critic network, thus avoiding querying the critic with the actions not in the offline dataset. For policy update, IQL use a weighted BC formulation

$$\max_\theta \mathbb{E}_{(s,a) \sim \mathcal{D}_{\text{offline}}} w(s, a) \cdot \log \pi_\theta(a|s), \tag{3}$$

where $w(s, a)$ denotes a data dependent weight, typically calculated based on the estimated advantages (Nair et al., 2020; Xu et al., 2022; Kostrikov et al., 2022).

## 3 OFFLINE AND ONLINE RL REVISITED: CONNECTIONS AND GAPS

**Connections between Offline and Online RL.** Offline RL and online RL are closely connected in many aspects. Historically, many offline RL approaches are branched off from off-policy RL algorithms to the full offline setting (Fujimoto et al., 2019; Fujimoto & Gu, 2021). Algorithms more specific to offline setting are then further developed, motivated by the challenges residing in the offline setting (Levine et al., 2020; Kumar et al., 2020). Besides algorithmic connections, offline and online RL are also complementary to each other in terms of strengths and weaknesses. Offline RL is sample efficient since no online interactions are required, but the performance is bounded by the fixed data set. Online RL enjoys more opportunities for performance improvement, but is comparatively much less sample efficient. Because of the connections and complementary strengths, instead of treating them as two isolated topics, it is more natural to connect both in pursuit of a performant policy in practice.

**The Direct Offline-Online Approach.** Because of the above mentioned connections, it is tempting to directly use the same algorithm (thus the same network architectures as well) for both phases. Unfortunately, this is ineffective in *either directions*: either directly using offline RL algorithms for online learning (*forward* direction), or directly using online RL algorithms for offline learning (*reverse* direction). The reverse direction has been explored extensively in the offline RL community. The current wisdom is that instead of directly using an existing online RL algorithm (e.g. TD3 (Fujimoto et al., 2018)), special treatments need to be incorporated into the algorithm (e.g. incorporating BC into TD3 (Fujimoto & Gu, 2021)) for handling the challenges arising in offline learning, due to issues such as querying the critic function with out-of-distribution actions.

In the *forward* direction, as noted in previous work (Nair et al., 2020), it is exceptionally difficult to first train a policy using offline data and then further improve it using online RL. There are some efforts in the literature on directly transferring the parameters learned offline for online fine-tuning, *i.e.*, by initializing the policy in Eqn.(1) with parameters learned offline. This scheme is illustrated in Figure 1 as the *Direct* approach for Offline-to-Online RL. While simple, this approach has several potential issues as noted in the literature. For example, one common issue is that the behavior of the offline policy can be compromised or even destroyed at the initial phase of online training, e.g. because of the noisy gradient for policy update due to cold start learning of critic network (Uchendu et al., 2022) (as in the case of reward-free pre-training) or distribution shift between offline and

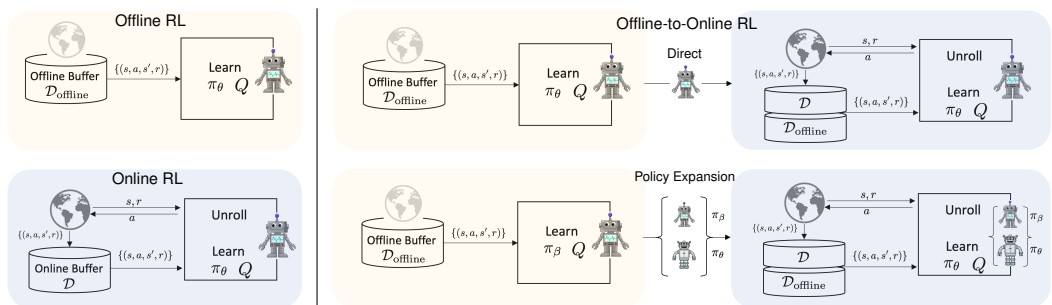

Figure 1: **Illustration of Different Training Schemes**. Offline training and online RL have been developed within their own training stages. Direct Offline-Online learning approach continues the online training stage after the offline stage is finished, updating the same policy network. The proposed Policy Expansion approach bridges offline and online training by retaining the policy after offline learning ($\pi_\beta$), and expand the policy set with another learnable policy ($\pi_\theta$) for capturing further performance improvements. The two policies both participate in interactions with environment and learning in an adaptive fashion.

online dataset (Lee et al., 2021). Another issue is the conservativeness of the offline RL algorithms. While this is a desirable feature when considering only offline training, it is not preferred for online learning, where exploration is valuable for a further improvement (Rezaeifar et al., 2022). This is a phenomenon that is commonly observed and reported in the literature (Lee et al., 2021; Campos et al., 2021; Uchendu et al., 2022).

## 4 BRIDGING OFFLINE AND ONLINE RL VIA POLICY EXPANSION

In this section, we will introduce a simple scheme called Policy Expansion for bridging offline and online training. It is worthwhile to note that the proposed scheme is orthogonal to specific off/online RL algorithms and is compatible to be used with different value-based offline and online algorithms. The final performance of such a combination may vary depend on the selection of methods.

### 4.1 POLICY EXPANSION AND ADAPTIVE COMPOSITION

**Policy Expansion.** To mitigate the above mentioned issues, we propose an alternative scheme that can be readily combined with existing algorithms. The proposed approach is illustrated in Figure 1. Given a policy $\pi_\beta$ obtained from offline training phase, instead of directly fine-tuning the parameters, we freeze $\pi_\beta$ and add it into a policy set $\Pi = [\pi_\beta]$. To enable further learning, instead of directly modifying $\pi_\beta$ as in the *Direct* method, which has the potential of destroying useful behaviors learned offline, we freeze $\pi_\beta$ and expand the policy set $\Pi$ with another learnable policy $\pi_\theta$ as

$$\Pi = [\pi_\beta, \pi_\theta] \tag{4}$$

which is responsible for a further performance improvement during online training. We refer this type of policy construction as Policy Expansion (PEX). It is intuitive to understand that the behavior of offline policy $\pi_\beta$ is free from being negatively impacted, while the newly added policy can be updated. The policies in the policy set $\Pi$ will all get involved into exploration and learning in an collaborative and adaptive manner as detailed in the following.

**Adaptive Policy Composition.** Both policies in the policy set $\Pi$ will form a single composite policy $\tilde{\pi}$, which will be used in both exploration and learning. More specifically, given the current state $s$, we first sample actions for each member of the policy set $\Pi$ and form a proposal actions set $\mathbb{A} = \{a_i \sim \pi_i(s) | \pi_i \in \Pi\}$. Then all the action proposals will be taken into consideration and they will be selected with the probability related to their potential utilities (e.g. values). For example, we can compute their values at the current state $\mathbf{Q}_\phi = [Q_\phi(s, a_i) | a_i \in \mathbb{A}] \in \mathbb{R}^K$, with $K$ denotes the cardinality of $\Pi$ (here $K = 2$), and construct a categorical distribution for selecting the final action:

$$P_\mathbf{w}[i] = \frac{\exp(Q_\phi(s, a_i)/\alpha)}{\sum_j \exp(Q_\phi(s, a_j)/\alpha)}, \quad \forall i \in [1, \cdots K] \tag{5}$$

where $\alpha$ is temperature. Then we can sample $\mathbf{w}$ from it for selecting the actions $\mathbf{w} \sim P_\mathbf{w}$ to decide which action will be used during unroll for interacting with environment. Using value for policy composition has been used in different contexts in the literature (Yu et al., 2021; Shah et al., 2022).

Conceptually, the composite policy $\tilde{\pi}$ can be represented as follows:

$$\tilde{\pi}(a|s) = [\delta_{a\sim\pi_\beta(s)}, \delta_{a\sim\pi_\theta(s)}]\mathbf{w}, \quad \mathbf{w} \sim P_{\mathbf{w}} \tag{6}$$

where $\mathbf{w} \in \mathbb{R}^K$ a one-hot vector, indicating the policy that is selected for the current state $s$. $\delta_{a\sim\pi}$ denotes the Dirac delta distribution centered at $a$ which is sampled from $\pi$.

By allowing only the newly added policy ($\pi_\theta$ in this case) to be fine-tuned while freezing all others ($\pi_\beta$), we can avoid the problem of compromising (e.g. destroying or forgetting) the behavior of offline policy. At the same time, we have the advantage of adaptiveness in the sense of allowing learning of new abilities. From this perspective, the policy expansion plays the role of bridging the offline and online learning phases, while mitigating commonly encountered issues. It is interesting to note that a similar compositional form of policy has appeared in DAgger (Ross et al., 2011), although with a uniform weight across states and under a different context of imitation learning. Here our compositional weight is state adaptive and the compositional policy is used for bridging offline-to-online reinforcement learning.

PEX has several advantages compared to direct offline-online learning (illustrated in Figure 1):

1. *offline policy preservation*: it can retain the useful behaviors learned during offline training phase by retaining the policy and avoid it being destroyed in the initial online training phase;
2. *flexibility in policy form*: the offline policy does not need to be of the same form with the online policy (e.g. same network structure) as in the direct offline-online approach, offering more flexibilities in design;
3. *adaptive behavior selection*: both the behavior of the offline policy and the online learning policy are used in interacting with the environment and they are involved in an adaptive manner, e.g., according to their respective expertise in handling different states.

---

**Algorithm 1** PEX: Policy Expansion for Offline-to-Online RL

---

**Input:** offline RL algorithm $\{L_{\text{offline}}^{Q_\phi}, L_{\text{offline}}^{\pi_\beta}\}$, online RL algorithm $\{L_{\text{online}}^{Q_\phi}, L_{\text{online}}^{\pi_\theta}\}^1$
**Initialize:** network parameters $\phi, \beta, \theta$, offline replay buffer $\mathcal{D}_{\text{offline}}$
**while** in *offline training phase* **do**
    % offline policy training using batches from the offline replay buffer $\mathcal{D}_{\text{offline}}$
    $\phi \leftarrow \phi - \lambda_Q \nabla_\phi L_{\text{offline}}^Q(\phi), \quad \beta \leftarrow \beta - \lambda_\pi \nabla_\beta L_{\text{offline}}^{\pi_\beta}(\beta)$
**end while**
Policy Expansion: $\tilde{\pi} = [\pi_\beta, \pi_\theta]$; transfer $Q_\phi$
**while** in *online training phase* **do**
    **for** each environment step **do**
        $a_t \sim \tilde{\pi}(a_t|s_t)$ according to (6), $s_{t+1} \sim T(s_{t+1}|s_t, a_t)$, $\mathcal{D} \leftarrow \mathcal{D} \cup \{(s_t, a_t, r(s_t, a_t), s_{t+1})\}$
    **end for**
    **for** each gradient step **do**
        % online training using batches from both $\mathcal{D}_{\text{offline}}$ and $\mathcal{D}$
        $\phi \leftarrow \phi - \lambda_Q \nabla_\phi L_{\text{online}}^Q(\phi), \quad \theta \leftarrow \theta - \lambda_\pi \nabla_\theta L_{\text{online}}^{\pi_\theta}(\theta)$
    **end for**
**end while**

---

### 4.2 BRIDGED OFFLINE-ONLINE TRAINING WITH POLICY EXPANSION

We focus on value-based RL algorithms for both stages in this work. For *offline-training*, we conduct offline RL with an offline RL algorithm (e.g. IQL) on the offline dataset to obtain the offline policy $\pi_\beta$. Then we can construct policy expansion following Eqn.(4) before entering the online phase. And we also transfer the Q function (critic) learned in the offline stage to online stage for further learning. We also transfer the offline buffer to online stage as an additional buffer, as shown in Figure 1.

For *online training*, we use the policy adaptively composed from the policy set as in Eqn.(6), and then conduct online training by interleaving environmental interaction and gradient update. The newly collected transitions are stored into the online replay buffer $\mathcal{D}$. For training, batches randomly sampled from both $\mathcal{D}$ and $\mathcal{D}_{\text{offline}}$ are used. The value loss and policy loss are calculated based on the losses corresponding to the chosen algorithm. The proposed scheme can be used together with different existing RL algorithms. The complete procedure is summarized in Algorithm 1, taking an offline RL and online RL algorithm as inputs.

---

[1] We represent a value-based RL algorithm succinctly with a pair of value and policy losses as $\{L^Q, L^\pi\}$.

## 5 RELATED WORK

**Pre-Training in RL.** A number of different directions have been explored in RL pre-training, including representation pre-training and policy pre-training. Note that for RL, pre-training can be either offline or online. Representative works include pre-training of the feature representation using standard representation learning methods (e.g. contrastive learning (Yang & Nachum, 2021)), dynamics learning-based representation learning (e.g. Schwarzer et al., 2021; Seo et al., 2022), unsupervised RL driven by intrinsic rewards in reward-free environments (e.g. Liu & Abbeel, 2021), or directly using ImageNet pre-training for visual RL tasks (e.g. Shah & Kumar, 2021; Yuan et al., 2022). Apart from representation pre-training, another category of work is on policy pre-training, with the goal of acquiring behaviors during the pre-training phase that are useful for the online phase. When the downstream task is unknown, there are approaches for unsupervised pre-training, e.g., maximizing behavior diversity (Eysenbach et al., 2019) or converting the action space Singh et al. (2021), with the hope of discovering some behaviors that are useful for the downstream task. When the offline-online task is more aligned, there are some early attempts on directly transferring policy parameters (Rajeswaran et al., 2018), based on the intuition that a policy initialized this way can produce more meaningful behaviors than randomly initialized networks. Some recent work focuses on behavior transferring, *i.e.*, leveraging the offline trained policy for exploration during online training (Campos et al., 2021; Uchendu et al., 2022). Our work falls into this latter category of methods. One notable difference compared to Campos et al. (2021); Uchendu et al. (2022) is that for the proposed approach, the offline policy is one part of the final policy, and with its role been determined adaptively.

**Data-Driven RL and Offline RL.** Training a policy by leveraging a large amount of existing data (*i.e.* data-driven RL) is a promising approach that is valuable to many real-world scenarios, where offline data is abundant. Offline RL is one active topic towards this direction. The main motivation of offline RL is to train a policy by leveraging a pre-collected dataset, without requiring additional environmental interactions (Levine et al., 2020). Many research works in this direction focus on addressing the special challenges brought by offline learning, including out-of-distribution value issue (Kumar et al., 2020). Common strategies include constraining the value (Kumar et al., 2020) to be small for out of distribution actions or the policy to be close to the action distribution of dataset (Fujimoto et al., 2019; Fujimoto & Gu, 2021). Recently, Kostrikov et al. (2022) proposes an implicit Q-learning (IQL) method as an alternative way to handle this issue. It learns a value function that predicts a certain expectile of the values for state-action pairs from the dataset, which can be used for computing the value target without querying the critic with out-of-distribution actions.

**Offline Training with Online Fine-tuning.** Combining offline data with online learning is an effective approach that have been demonstrated by several early attempts with demonstration data (Vecerík et al., 2017; Hester et al., 2018; Nair et al., 2018; Rajeswaran et al., 2018). Traditionally, the offline RL methods are purely focused on the offline training setting. However, the offline learned policy could be limited in performance given a fixed dataset. In the case when further online interaction is allowed, it would be natural to fine-tune the policy further with data collected online. This paradigm of two-stage policy training is related to the iterative interleaved policy and data collection schemes used in imitation learning (Ross et al., 2011; Ross & Bagnell, 2012). Several different approaches have been explored towards online fine-tuning of a offline pre-trained policy, including balancing offline-online replay data (Lee et al., 2021), parameter transferring (Rajeswaran et al., 2018; Xie et al., 2021), policy regularization (Rudner et al., 2021; Tirumala et al., 2020) and guided exploration (Campos et al., 2021; Uchendu et al., 2022). It has been observed that directly applying some offline RL methods does not benefit from online interactions (Nair et al., 2020; Uchendu et al., 2022), potentially due to the conservative nature of the offline policy (Nair et al., 2020). Based on this observation, there are some recent efforts on developing algorithms that are not only suitable for offline training, but also can leverage online interactions for further learning. Nair et al. (2020) shows that an advantage-weighted form of actor-critic method is suitable for this purpose. IQL (Kostrikov et al., 2022) also leverages a similar form for policy learning and shows that it can benefit from online fine-tuning. Our work falls into this category and we provide an alternative approach for leveraging offline pre-training for helping with online training.

## 6 EXPERIMENTS

In this section, we first evaluate the effectiveness of the proposed approach on various types of benchmark tasks with comparison to a number of baseline methods. Then we further show a number of extensions where the proposed approach can also be applied.

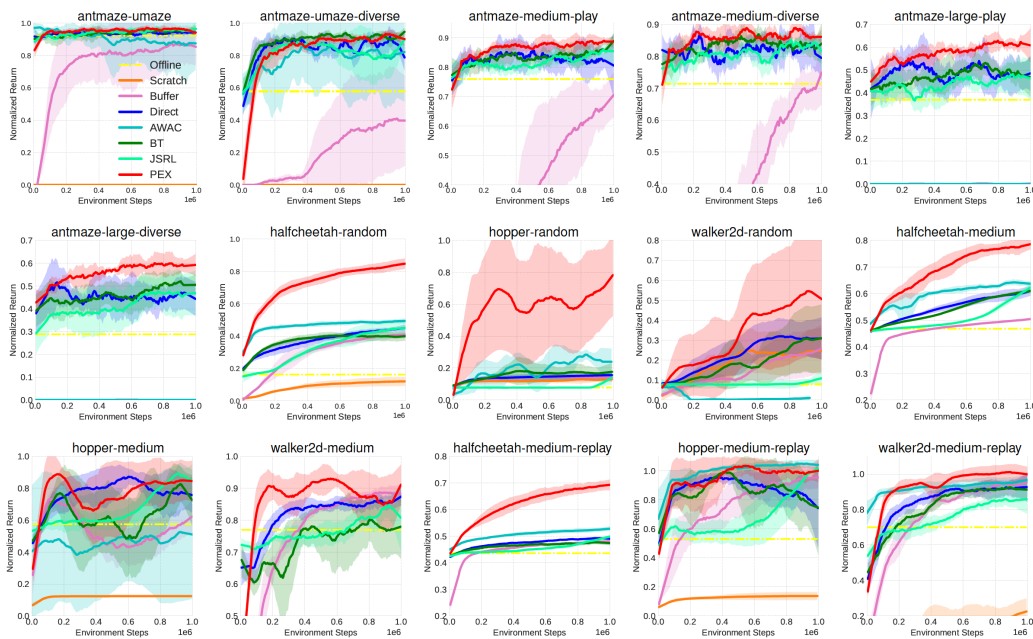

Figure 2: **Normalized Return Curves** of different methods on benchmark tasks from D4RL (Fu et al., 2020). IQL is used as the backbone apart from the AWAC baseline (Nair et al., 2020).

## 6.1 OFFLINE-TO-ONLINE RL EXPERIMENTS

**Tasks and Settings.** We use the standard D4RL benchmark which has been widely used in offline RL community (Fu et al., 2020). For offline learning, we use the provided dataset for training. For online learning, we use the accompanied simulator for interaction and training. For offline phase, 1M training steps are used for training. Then we run online fine-tuning for another 1M environmental steps. Here we use IQL (Kostrikov et al., 2022) as the backbone algorithm for all methods listed below. The training is repeated with 5 different random seeds.

**Baselines.** We compare the proposed approach with the following baselines: *(i)* `Offline`: offline training using IQL, without online fine-tuning; *(ii)* `Scratch`: train IQL online from scratch, without offline-pre-training. *(iii)* `Buffer` (Vecerík et al., 2017): train IQL online without offline pre-training, but has access to the offline buffer during online training, *i.e.* using a buffer as $\mathcal{D} \cup \mathcal{D}_{\text{offline}}$. *(iv)* `Direct` (Kostrikov et al., 2022): a direct offline-to-online approach by directly transferring parameters trained offline to online stage using IQL (Kostrikov et al., 2022), which is a recent and representative RL algorithm that shows state-of-the-art performance on offline RL while allows online fine-tuning; *(v)* `AWAC` (Nair et al., 2020): an approach that uses an advantage-weighted form of actor-critic method for offline-to-online RL; *(vi)* `Off2On` (Lee et al., 2021): a recent offline-to-online RL method that uses an ensemble of offline trained value and policies together with a balanced offline-online replay scheme; *(vii)* `BT` (Campos et al., 2021): Behavior Transfer which is an approach that leverages an offline learned policy in exploration, where the offline policy is used for exploration for a consecutive number of steps sampled from a distribution once activated; *(viii)* `JSRL` (Uchendu et al., 2022): Jump Start RL, which divides the rollout of a trajectory into two parts, using the offline learned policy for the first part and then unrolling with the online learning policy for the rest of trajectory. `PEX` denotes the proposed approach, which has the same offline and online RL algorithms as `Direct`, and is only different in using *Policy Expansion* for connecting the two stages. More resources are available on the project page. [2]

The return curves for all the tasks are shown in Figure 2. The aggregated return across all tasks is shown in Figure 3. The returns are first averaged across task and then across runs. It can be observed that all methods show some improvements after online training in general, compared to the initial performance before online training. `Scratch` has

Figure 3: **Aggregated Return Curves** across tasks (IQL-based).

[2] https://sites.google.com/site/hczhang1/projects/pex

the lowest overall performance across all tasks. On the challenging sparse reward antmaze tasks, **Scratch** cannot learn a meaningful policy at all. **Buffer** has better performance than **Scratch** when incorporating the offline buffer, indicating that the offline buffer has some benefits in helping with learning. Notably, on the previously zero-performance antmaze tasks, **Buffer** achieves reasonable performance with the help of the offline buffer, leading to a large overall improvement over **Scratch**, (*c.f.* Figure 3). **Direct** (IQL-based) shows large improvements over **Offline** on average as shown in Figure 3, implying the benefits brought by the the additional online training over pure offline training. **Off2On** also shows large improvement during fine-tuning and achieves strong overall performance (Figure 3). **BT** shows some improvements over **IQL** on some tasks such as `antmaze-medium-play` and `antmaze-large-diverse`, with an overall performance comparable to that of **Direct**. **JSRL** outperforms **Direct** and **BT** on some tasks (e.g. `hopper-medium`, `hopper-medium-replay`), potentially due to its different way of leveraging the offline policy, and its overall performance is similar to **BT**. The proposed **PEX** approach performs comparably to other baselines on some tasks while outperforming all baseline methods on most of the other tasks (*c.f.* Figure 2), and outperforms baselines methods overall (*c.f.* Figure 3), demonstrating its effectiveness.

## 6.2 HETEROGENEOUS OFFLINE-ONLINE RL BRIDGING VIA POLICY-EXPANSION

We have shown the application of **PEX** to the case where both the offline and online algorithms are the same (referred to as **PEX-IQL** here since both are IQL) in Section 6.1. In this section, we further show the applicability of the proposed scheme in bridging heterogeneous RL algorithms, *i.e.* different RL methods are used for the offline and online stages. As an example, here we use IQL (Kostrikov et al., 2022) and SAC (Haarnoja et al., 2018) for offline and online RL stages respectively. We compare with **Scratch** (vanilla SAC (Haarnoja et al., 2018)), **Buffer** (SAC with additional offline replay buffer) as well as **Direct** (directly transferring policy and critic parameters learned with offline IQL to SAC for further online learning). Again **PEX** uses the same offline and online algorithms as in **Direct** but uses policy expansion instead of the direct transferring approach. The normalized return curves aggregated across tasks are shown in Figure 4.

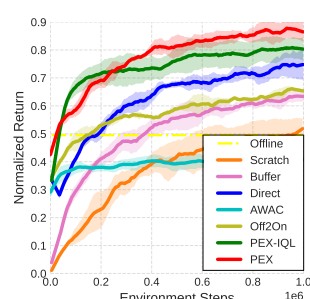

Figure 4: **Aggregated Return Curves** across benchmark tasks (SAC-based).

Individual return curves are shown in Appendix A.8. It can be observed that there is an overall improvement by simply applying **PEX** to the heterogeneous offline-online RL setting as well.

## 6.3 ABLATION STUDIES

We will inspect the impact of several factors on the performance of the proposed method in the sequel.

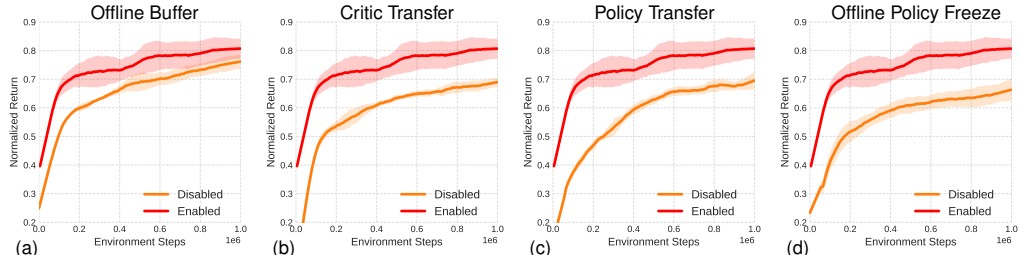

Figure 5: **Ablation Results** on a number of factors. Orange curves correspond to ablation variants.

**Offline-Buffer.** This experiment investigates the impacts of including the offline replay buffer in online training stage. The results are shown in Figure 5(a). As can be observed, the inclusion of the offline replay buffer helps with the performance, but the performance drop caused by disabling offline replay buffer is smaller than the change of other algorithmic components (*c.f.* Figure 5(b)~(d)).

**Critic Transfer.** This experiment studies the impacts of transferring the critic parameters trained from offline to online stage. As can be observed from Figure 5(b), disabling critic transferring will also greatly decrease the performance both in terms of sample efficiency as well as final performance.

**Policy Transfer.** This experiment examines the impacts of transferring the offline pre-trained policy in the online training stage. When policy transferring is disabled, there is no need to use policy

expansion in the online training stage. The results are shown in Figure 5(c). It can be observed that there is a clear performance drop when policy transferring is disabled.

**Offline Policy Freeze.** The offline learned $\pi_\beta$ is freezed during the online learning stage in **PEX**. This experiment investigates the impact of this factor. If disabled, $\pi_\beta$ will be trained in the same way as $\pi_\theta$ during online learning. It is observed from Figure 5(d) that freezing the offline policy is important. Training by disabling policy freezing has a clear performance drop. This is consistent with the intuition on the usefulness of offline policy preservation, which is one advantage of our approach. Another set of ablation results are deferred to Appendix A.4.

## 6.4 VISUALIZATION AND ANALYSIS

**Policy Composition Probability.** Since a composite policy is involved in the proposed approach, it is interesting to inspect the participation of each member policy from the policy set $\Pi = \{\pi_\beta, \pi_\theta\}$ when interacting with the environment. We visualize the policy compositional probability $P_\mathbf{w}$ during the rollout within a trajectory after training, as shown in Figure 6. It can be observed that composition probability for each member policy is state-adaptive and is changing along the progress of a trajectory, implying that both policies contribute to the final policy in an adaptive manner.

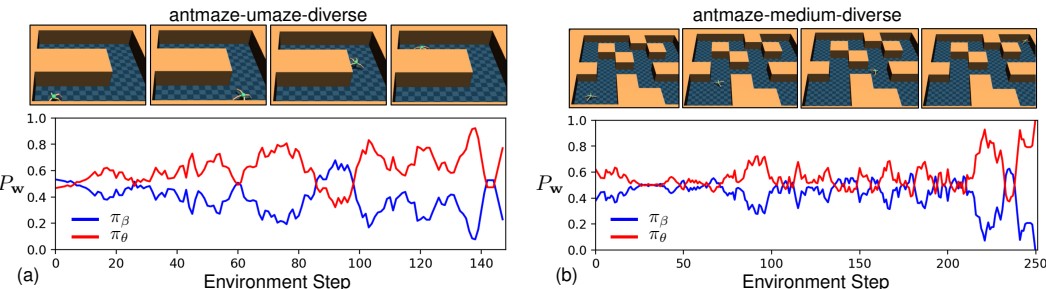

Figure 6: **Visualization of Policy Composition Probability** during the rollout of one trajectory. The probability curves are smoothed for visualization purpose.

**State Space Associations of Member Policies.** To get a better understand of the role of the member policies in the policy set, we visualize the association between the states and its selected policy. For this purpose, we embed a set of states into a 2D space using t-SNE (van der Maaten & Hinton, 2008), and then visualize the association of offline policy $\pi_\beta$ and the newly expanded policy $\pi_\theta$ to states in the projected space. States that select the offline policy $\pi_\beta$ are colored with blue and states that select $\pi_\theta$ are colored with red. It can be observed that $\pi_\theta$ and $\pi_\theta$ cover different parts of the state space, indicating that they have some complementary functionalities and are preferred differently at different states.

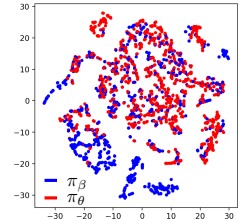

## 7 CONCLUSIONS, LIMITATIONS AND FUTURE WORK

We highlight the usefulness of properly connecting offline and online stages of reinforcement learning to gain the benefits of both worlds and present a policy expansion scheme as an attempt toward this direction. This scheme is an instance in the direction orthogonal to developing completely new offline-to-online RL algorithms. The proposed approach is simple and can be combined with existing RL algorithms, and is illustrated with two different combinations in this work. Experiments demonstrate the effectiveness of the proposed approach on a number of benchmark tasks.

While achieving promising performance, the proposed approach also has some limitations. One limitation is that the number of parameters grows with the number of policies in the policy set. While this might not be an issue when the policy set is small as the case in this work, it will be less parameter efficient in the presence of a large policy set. One possible way to address this issue is by introducing a distillation stage (Rusu et al., 2016), by consolidating the multiple policies into a network with a smaller number of parameters. Generalizing the proposed scheme to the case with a set of pre-trained skill policies is an interesting direction (Eysenbach et al., 2019; Shu et al., 2018). For offline learning, we have built upon the strong IQL method. It would be interesting to see how much we can gain by upgrading it with more recently developed offline RL methods together with different online methods. The idea of using a policy set itself can potentially be applied to other cases beyond offline to online RL. We leave the exploration of its generalization and application as an interesting future work.

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

## A  APPENDIX

### A.1  DETAILS ON BASELINES

In this section, we provide more details on the baseline methods.

**Behavior Transfer (`BT`).** This approach is originally proposed for the case of pre-training using unsupervised reinforcement learning and then transfers the learned behaviors for later learning. It is also originally used in discrete action cases. Here we adapt this idea to build a baseline in our setting based on the approach of transferring behaviors in (Campos et al., 2021). Following (Campos et al., 2021), a Zeta distribution with parameter $a = 2$ is used for determining the number of steps of persistent unroll using the offline policy. The number of persistent unroll steps is re-sampled when currently not in the middle of a persistent unroll and with a random number sampled from $[0, 1]$ is smaller than a threshold $\epsilon$. $\epsilon = 0.1$ is used in experiments.

**Jump Start RL (`JSRL`).** `JSRL` divides the complete unroll of one trajectory into two parts, with the first part obtained with the offline policy $\pi_\beta$, while the second part is obtained with the current learning policy $\pi_\theta$. The core idea of `JSRL` is to initially make the first part longer, and then progressively reducing the length of the first part, thus forming a type of curriculum to help with exploration and learning. We follow this core idea by using a linear scheduler that anneals from the max episode length to 0 for progressively decreasing the first stage of unroll using the prior policy $\pi_\beta$.

**Offline-to-Online RL (`Off2On`).** `Off2On` is an approach that aims to improve offline to online RL performance (Lee et al., 2021). It incorporates two components to cope with the challenges of offline-to-online RL: 1) an ensemble of critic networks and policy networks are trained during offline training stage, in contrast to typical offline RL, which only learns one policy network and critic network. [3] Following the setting in Lee et al. (2021), ensemble size of 5 is used. 2) balanced replay: a balanced offline-online replay scheme to properly trade off between the usage of offline and online samples. We used the code released by the authors of (Lee et al., 2021). [4]

### A.2 IMPLEMENTATION DETAILS ON POLICY EXPANSION

**Policy Expansion (`PEX` with IQL-backbone).** The proposed scheme can be combined with many different existing RL methods. Here we illustrate with an concrete instantiation using IQL (Kostrikov et al., 2022). For pre-training, we conduct offline RL with IQL on the offline dataset to obtain the offline policy $\pi_\beta$. Then we conduct policy expansion as Eqn.(6) before switching to online phase.

For *offline training*, we follow IQL (Kostrikov et al., 2022) to train the parameters of the value function using the TD-learning approach as follows:

$$L_Q(\phi) = \mathbb{E}_{(s,a,r,s')\sim\mathcal{D}_{\text{offline}}}\|Q_\phi(s,a) - (r + \gamma V(s'))\|^2 \tag{7}$$

$$L_V(\psi) = \mathbb{E}_{(s,a)\sim\mathcal{D}_{\text{offline}}} L_2^\tau[Q_{\bar{\phi}}(s,a) - V_\psi(s)] \tag{8}$$

where $L_2^\tau(u) = |\tau - \mathbb{1}(u < 0)|u^2$ (Kostrikov et al., 2022), with $\tau$ denotes the expectile value (Kostrikov et al., 2022). $\bar{\phi}$ denotes a set of target parameters that are periodically copied from $\phi$. The policy network $\pi_\theta$ can be further updated by leveraging the updated critic network. Here we again follow IQL (Kostrikov et al., 2022) and use the weighted form for actor training:

$$L_\pi(\beta) = \mathbb{E}_{(s,a)\sim\mathcal{D}_{\text{offline}}} - \lfloor w(s,a) \rfloor \cdot \log \pi_\beta(a|s), \tag{9}$$

with $w(s,a) = \exp((Q(s,a) - V(s))/\alpha)$ and $\lfloor \cdot \rfloor$ denotes the gradient stopping operator.

For *online training*, we first expand the policy set as in Eqn.(4), and then conduct online training by interleaving environmental interaction and gradient update. The newly collected transitions are saved into replay buffer $\mathcal{D}$. For training, batches randomly sampled from both $\mathcal{D}$ and $\mathcal{D}_{\text{offline}}$ are used. The values losses are the same as the offline phase, and the policy loss is computed against $\pi_\theta$:

$$L_\pi(\theta) = \mathbb{E}_{s\sim\mathcal{D}_{\text{offline}}\cup\mathcal{D},a\sim\tilde{\pi}(s)} - \lfloor w(s,a) \rfloor \cdot \log \pi_\theta(a|s), \tag{10}$$

For $a_0 \sim \pi_\beta(s_t)$, in practice, we simply take the greedy action from the policy. Following Kostrikov et al. (2022), we model the policy using a non-squashed Gaussian distribution with mean squashed to action range, and use state independent standard deviations.

**Policy Expansion (`PEX` with with SAC as online backbone).** The main procedure of online training is the same as SAC (Haarnoja et al., 2018), with an adaptation for the actor training by changing the part of the loss based on Q-function from $\mathbb{E}_{s\sim\mathcal{D}}\mathbb{E}_{a\sim\pi_\theta} \left\| \lfloor \frac{\partial Q(s,a)}{\partial a} + a \rfloor - a \right\|^2$ (which is an equivalent form of loss to $\mathbb{E}_{s\sim\mathcal{D}}\mathbb{E}_{a\sim\pi_\theta} - Q(s,a)$) to $\mathbb{E}_{s\sim\mathcal{D}}\mathbb{E}_{a\sim\tilde{\pi},a_0\sim\pi_\theta} \left\| \lfloor \frac{\partial Q(s,a)}{\partial a} + a \rfloor - a_0 \right\|^2$.

---

[3]Here we refer the commonly used double critic replica to reduce value overestimation as in SAC (Haarnoja et al., 2018) and TD3 (Fujimoto et al., 2018) as a single critic network. The ensemble used in Lee et al. (2021) is applied on top of this.

[4]https://github.com/shlee94/Off2OnRL

## A.3 THEORETICAL JUSTIFICATION OF PEX

Here we provide some theoretical justification of the proposed form of adaptive policy compostional distribution $P_{\mathbf{w}}$ in Eqn.(5):

$$P_{\mathbf{w}}[i] = \frac{\exp(Q_\phi(s, a_i)/\alpha)}{\sum_j \exp(Q_\phi(s, a_j)/\alpha)}, \quad \forall i \in [1, \cdots K], \tag{11}$$

where $K$ denotes the number of actions and in our case $K = 2$.

$P_{\mathbf{w}}$ can be viewed as a discrete policy with action dimension of two, which selects between two candidate actions $a_1, a_2$, one from offline policy $\pi_\beta$ and the other from $\pi_\theta$, i.e., $a_1 \sim \pi_\beta$ and $a_2 \sim \pi_\theta$. Ideally, it assigns a higher probability to actions with higher values at the current state $s$. The definition of $P_{\mathbf{w}}$ essentially a reflection of this intuition, Actually, it can also be derived as shown below.

By definition, we have the following form for the compositional policy $\tilde{\pi}$:

$$\tilde{\pi}(a|s) = \int_{a_1, a_2} \pi_\beta(a_1|s)\pi_\theta(a_2|s) \sum_i P_{\mathbf{w}}(i|s, a_1, a_2)\delta(a = a_i)da_1 da_2 \tag{12}$$

which is essentially another form of Eqn.(6).

For V-value, we can have the following derivations

$$V(s) = \int_a \tilde{\pi}(a|s)Q_\phi(s, a)da \tag{13}$$

$$\text{(plug in the definition of policy in Eqn.(12))} \tag{14}$$

$$= \int_a \left[\int_{a_1, a_2} \pi_\beta(a_1|s)\pi_\theta(a_2|s) \sum_i P_{\mathbf{w}}(i|s, a_1, a_2)\delta(a = a_i)da_1 da_2\right] Q_\phi(s, a)da \tag{15}$$

$$= \int_{a_1, a_2} \pi_\beta(a_1|s)\pi_\theta(a_2|s) \sum_i P_{\mathbf{w}}(i|s, a_1, a_2) \left[\int_a \delta(a = a_i)Q_\phi(s, a)da\right] da_1 da_2 \tag{16}$$

$$= \int_{a_1, a_2} \pi_\beta(a_1|s)\pi_\theta(a_2|s) \sum_i P_{\mathbf{w}}(i|s, a_1, a_2)Q(s, a_i)da_1 da_2 \tag{17}$$

$$= \mathbb{E}_{a_1 \sim \pi_\beta, a_2 \sim \pi_\theta} \left[\sum_i P_{\mathbf{w}}(i|s, a_1, a_2)Q_\phi(s, a_i)\right] \tag{18}$$

Applying entropy regularization term for policy learning, we have

$$\mathbb{E}_{a_1 \sim \pi_\beta, a_2 \sim \pi_\theta} \left[\sum_i P_{\mathbf{w}}(i|s, a_1, a_2) \left(Q_\phi(s, a_i) - \alpha \log P_{\mathbf{w}}(i|s, a_1, a_2)\right)\right]. \tag{19}$$

Solving for $P_{\mathbf{w}}$ given $\pi_\beta$ and $\pi_\theta$, we have

$$P_{\mathbf{w}}(i|s, a_1, a_2) \propto \exp(Q_\phi(s, a_i)/\alpha), \quad \forall i \in [1, 2]. \tag{20}$$

Therefore, for the case of two actions, we have

$$P_{\mathbf{w}}(i|s, a_1, a_2) = \frac{\exp(Q_\phi(s, a_i)/\alpha)}{\sum_j \exp(Q_\phi(s, a_j)/\alpha)}, \quad \forall i \in [1, 2]. \tag{21}$$

## A.4 ADDITIONAL ABLATION RESULTS

We provide more ablations experiments and results in this section.

**Double Parameters.** Since there are two member policies in the proposed approach. A natural question to ask is whether properly doubling the number of parameters of a policy could achieve similar performance.

To verify this, we construct a type of policy network with (roughly) doubled number of parameters as $\pi_\theta(s) = g_{\theta_g} \circ f_{\theta_f}(s)$, where $f_{\theta_f}(s)$ encodes the observation $s$ into a feature vector and $g$ maps the feature vector to the action distribution. To (approximately) double the number of parameters, we implement $f_{\theta_f}$ as $f_{\theta_f}(s) \triangleq f_{\theta_1}(s) + f_{\theta_2}(s)$. $f_{\theta_1}$ is initialized by transferring parameters $\theta$ learned in the pre-train stage. The aggregated return curves are shown on the right, for doubling the number of policy parameters as described above (**Double-Param**), and the proposed approach. It can be observed that by comparison that simply increasing the number of parameters cannot help much with the performance, implying the importance of the proposed structure in **PEX**.

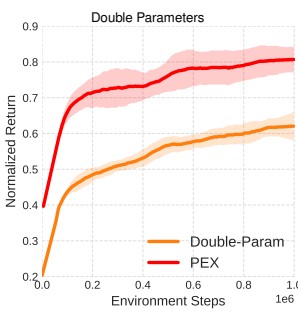

### A.5 MORE DETAILS ON TASKS

We summarized the tasks used for experiments and their corresponding D4RL environment names in Table 1.

| Task | D4RL Environment Name |
|---|---|
| antmaze-umaze | antmaze-umaze-v0 |
| antmaze-umaze-diverse | antmaze-umaze-diverse-v0 |
| antmaze-medium-play | antmaze-medium-play-v0 |
| antmaze-medium-diverse | antmaze-medium-diverse-v0 |
| antmaze-large-play | antmaze-large-play-v0 |
| antmaze-large-diverse | antmaze-large-diverse-v0 |
| halfcheetah-random | halfcheetah-random-v2 |
| hopper-random | hopper-random-v2 |
| walker-random | walker-random-v2 |
| halfcheetah-medium | halfcheetah-medium-v2 |
| hopper-medium | hopper-medium-v2 |
| walker-medium | walker-medium-v2 |
| halfcheetah-medium-replay | halfcheetah-medium-replay-v2 |
| hopper-medium-replay | hopper-medium-replay-v2 |
| walker-medium-replay | walker-medium-replay-v2 |

Table 1: **Tasks and the corresponding environment names in D4RL**.

### A.6 HYPER-PARAMETERS

The hyper-parameters and values are summarized in Table 2. There are two algorithmic related hyper-parameters inherited from IQL since we instantiate PEX based on it: temperature $\alpha$ (corresponding to $\beta^{-1}$ in IQL paper (Kostrikov et al., 2022)) and expectile value $\tau$. The rest of the hyper-parameters are common ones in RL algorithms, including for example batch size, network structure and size, target update etc. We use the hyper-parameter values from the IQL paper (Kostrikov et al., 2022) in our experiments.

### A.7 HYPER-PARAMETER ABLATIONS

**Impact of $\alpha$.** $\alpha$ scales the Q values before constructing $P_{\mathbf{w}}$:

$$P_{\mathbf{w}}[i] = \frac{\exp(\alpha^{-1}Q_\phi(s, a_i))}{\sum_j \exp(\alpha^{-1}Q_\phi(s, a_j))}, \quad \forall i \in [1, \cdots K]. \tag{22}$$

We show the impacts of its value on performance in Figure 7. We have use a fixed value ($\alpha^{-1} = 10$ for all antmaze tasks and $\alpha^{-1} = 3$ for all locomotion tasks according to the IQL paper (Kostrikov et al., 2022) (the $\beta$ parameter of IQL). Here we show the impacts of different values for $\alpha^{-1}$ by setting the inverse temperature as $0.5\alpha^{-1}$, $1\alpha^{-1}$, $2\alpha^{-1}$. We note that other values are possible for different environments. While it is impractical to tune this parameter for each task, it is possible to make it more adaptive by auto-tuning in a way similar to the temperature turning scheme used in SAC (Haarnoja et al., 2018).

| Hyper-parameters | Values |
|---|---|
| number of parallel env | 1 |
| discount | 0.99 |
| replay buffer size | 1e6 |
| batch size | 256 |
| MLP hidden layer size | [256, 256] |
| learning rate | 3e-4 |
| initial collection steps | 5000 |
| target update speed | 5e-3 |
| expectile value $\tau$ | 0.9 (0.7) |
| inverse temperature $\alpha^{-1}$ | 10 (3) |
| number of offline iterations | 1M |
| number of online iterations | 1M |
| number of iteration per rollout step | 1 |
| target entropy (SAC) | $-d$ |

Table 2: Hyper-parameter values. Values in brackets are used for locomotion tasks. The rest of the hyper-parameter values are shared across all tasks. $d$ denotes the action dimension.

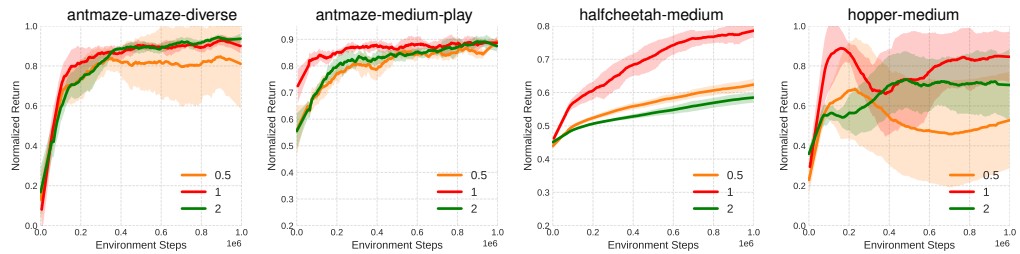

Figure 7: Impact of inverse temperature $\alpha^{-1}$. Results with inverse temperature as $0.5\alpha^{-1}$, $1\alpha^{-1}$, $2\alpha^{-1}$ respectively.

**Impact of Policy Entropy**. The entropy of $\pi_\theta$ is auto-tuned to match a target entropy (Haarnoja et al., 2018), which is empirically set as $e_{\text{target}} \triangleq -d$ where $d$ denotes the dimensionality of action. This is inherited from SAC and the same setting $(-d)$ is applied to all tasks. To show the impact of different target entropy values, we experiment with three different settings: $0.5e_{\text{target}}$, $1 \cdot e_{\text{target}}$, $2e_{\text{target}}$ on several tasks. The results are summarized in Figure 8. It can be observed that overly small or large target entropy values could decrease the performance on some tasks.

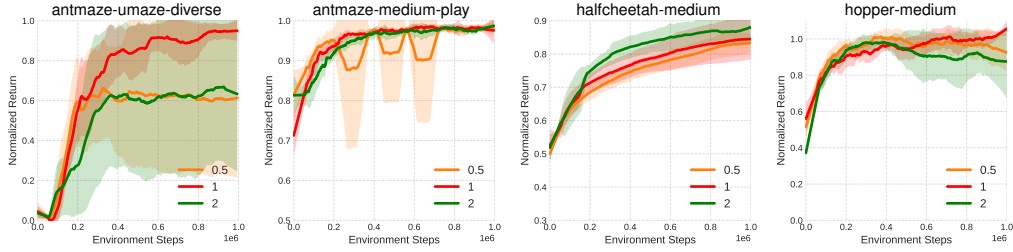

Figure 8: Impact of policy entropy (SAC-based online stage). Results with target entropy as $0.5e_{\text{target}}$, $1e_{\text{target}}$, $2e_{\text{target}}$ respectively.

### A.8 MORE RESULTS ON HETEROGENEOUS OFFLINE-ONLINE RL

Here we show detailed return curves for each task when using SAC (Haarnoja et al., 2018) as the backbone online RL algorithm. The results are shown in Figure 9.

### A.9 OFFLINE-ONLINE TRAINING WITH REWARD-FREE OFFLINE SETTING

We have shown using policy expansion for bridging offline IQL with online IQL/SAC in the main paper. Here we further demonstrate the applicability of the proposed approach by using BC as the

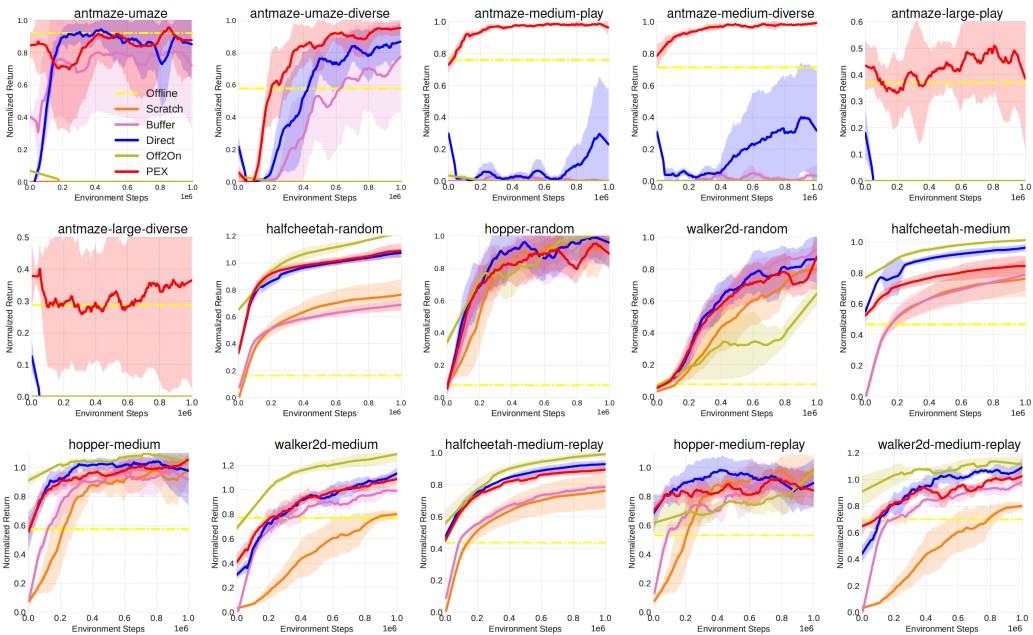

Figure 9: **Return Curves** of different methods using SAC (Haarnoja et al., 2018) as the online RL backbone algorithm, including the Off2On method (Lee et al., 2021). The returns are normalized according to D4RL benchmark (Fu et al., 2020).

offline learning module. We use IQL as the online module for illustration. In this section, we further demonstrate the applicability of the proposed approach to the setting with offline dataset without reward annotations. This is sometimes an encountered setting in practice, where plenty of offline data exist (e.g. from previous iteration of experiments or other agents), but without reward annotations. Since the offline dataset is only broadly related to the task and could potentially contain irrelevant or noisy data (*w.r.t.* the current task in consideration), direct behavior cloning could be sub-optimal.

Standard offline RL methods are inapplicable for offline pre-training in this case. We therefore use behavior cloning (**BC**) (Pomerleau, 1988), which is a simple and widely used approach for imitation learning. After offline pre-training using **BC**, the offline policy $\pi_\beta$ is obtained and transferred to the second stage of online training. In terms of Algorithm 1, only the policy loss $L_{\pi_\beta}$ is present in the offline training phase. The offline datasets not used in online training since it has no rewards.

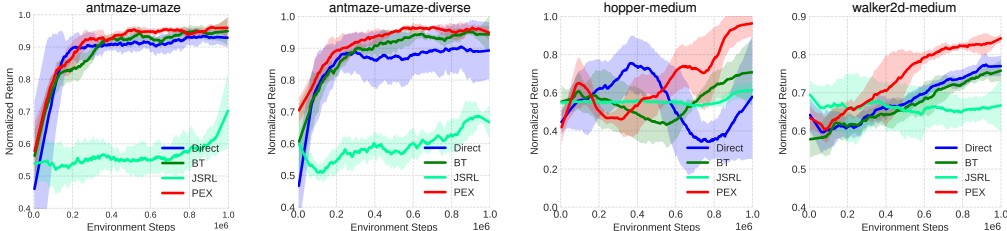

Figure 10: **Return Curves** for reward-free offline pre-training setting.

We use subset of tasks from D4RL (Fu et al., 2020) where **BC** has non-trivial performances. The results are shown in Figure 10. It can be observed that **JSRL**, which performs well under the case of having offline rewards, performs less well in the reward-free case. One potential reason is that it hinges on the offline policy to induce the state distribution (Uchendu et al., 2022). In the case where the offline policy is not highly performant, as the case here obtained from **BC**, its effectiveness could be compromised. **BT** performs comparable to or better than **Direct** online fine-tuning with IQL, potentially due to the fact that **BT** leverage the $\pi_\beta$ in a way that has a shorter commitment

length for each time the offline policy is selected. The proposed approach also shows similar online improvements comparable or sometimes better than the best performing methods.

## A.10 POLICY USAGE

We show the policy usage of $\pi_\beta$ and $\pi_\theta$ during online unroll along the process of one training session for **BT** and **JSRL** as well as **BC-PEX** (pre-trained with BC) and **IQL-PEX** (pre-trained with IQL) in Figure 11. The return curves of **BC-PEX** are shown in Figure 10. The corresponding return curves of **IQL-PEX** are shown in Figure 2. As can be observed, the usage of offline policy is changing along the course of training, and the dynamics of the usage evolution is different for different tasks.

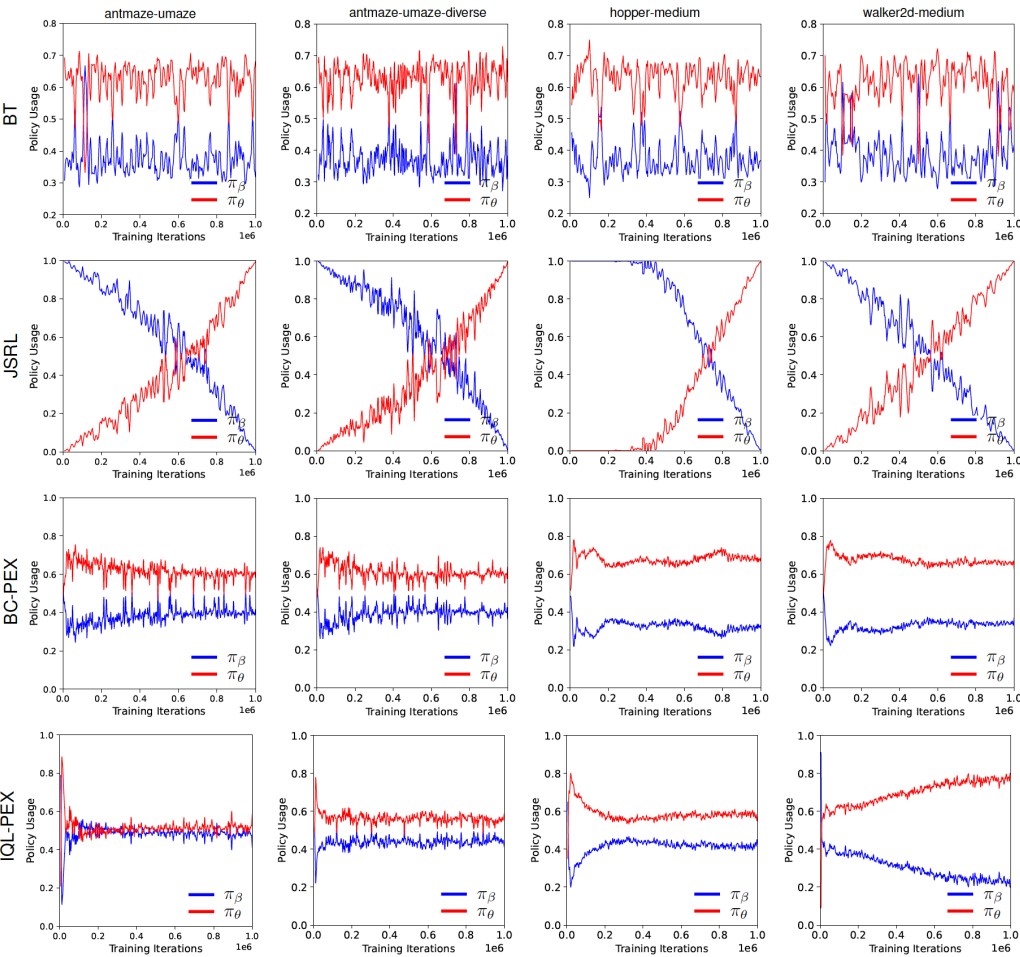

Figure 11: **Policy Usage** of $\pi_\beta$ and $\pi_\theta$ along the process of online training for different methods.

