# OpenReview forum: "Policy Expansion for Bridging Offline-to-Online Reinforcement Learning"
_ICLR.cc/2023/Conference — ICLR 2023 poster_

### Official Review · Reviewer_NB3z · 2022-10-17

**Confidence:** 4
**Correctness:** 4
**Technical Novelty And Significance:** 3
**Empirical Novelty And Significance:** 4
**Recommendation:** 8

**Clarity, Quality, Novelty And Reproducibility:**

**Clarity**
should be improved as stated above

**Quality**
Good

**Novelty**
Good

**Reproducibility**
should be improved as stated above


**Strength And Weaknesses:**

**Strengths**
- The paper deals with a problem that is relevant for practical applications
- The method is very interesting. I find it surprising that it works.

**Weaknesses**
- The, in my opinion, central aspect of how actions are selected is presented somewhat scarcely.  I am not sure that the method can be reliably re-implemented using the explanations provided.

**Further suggestions for improvement**
- I think the calculus of how the selection probabilities are calculated from the Q-values of the candidate actions should be explicitly presented.
- The temperature parameter $\alpha$ probably has central influence on the functioning of the method. It would be interesting to see this influence through experiments.
- Also the entropy of the two policies may have a significant influence. It would be interesting to see this influence through experiments.
- Some sentences could be worded more concisely. For example, sentence „After learning the offline policy, we use it as one candidate policy in a policy set, and further learn another policy that will be responsible for further learning as an expansion to the policy set" seems a bit awkward and thus difficult to understand.\
I stumbled over „limiting its applicability to many practical scenarios“, wouldn't „limiting its applicability in many practical scenarios“ or „which limits its applicability in many practical scenarios“ be better?
- Why do you write "e.g." in italics while "et al." is not italic? This seems inconsistent to me.

„to a a number“\
„onehot“ -> „one-hot“\
„the policy set set as“\
„q-learning“ -> „Q-learning“\
„Should i run“ -> „Should I run“


**Summary Of The Paper:**

The paper addresses how online RL can benefit from an existing dataset and/or a policy created by offline RL. The suggested method proposes that both the offline policy (fixed) and the online policy (to be trained) provide action candidates, which are then evaluated using a Q-function (to be learned online). The evaluations are converted into selection probabilities and the action to be executed is drawn accordingly. The method is being tested on several D4RL benchmark tasks.

**Summary Of The Review:**

A surprisingly simple approach to solve an important problem. The description of the approach should be more explicit in parts and thus more reproducible. The language should be improved in parts, errors should be fixed.

---

> ### Author Response · Authors · 2022-11-19
> **Response to Reviewer NB3z**
>
> We thank the reviewer for the positive review and insightful comments.
>
> **Weakness: The central aspect of how actions are selected is presented somewhat scarcely. I am not sure that the method can be reliably re-implemented using the explanations provided.**
>
> The proposed approach is indeed very simple and straightforward
>
> -   The construction of the compositional policy is provided in Eqn.(5) and Eqn.(6) and we have updated the calculation of $P_{\mathbf{w}}$ in Eqn.(5) with an explicit form.
>
>
> -   The training process is summarized in Algorithm 1, which mainly contains standard offline RL and online RL procedures, using the PEX policy;
>
>
> -   To facilitate fellow researchers in the community, we will release the code as well as pre-trained offline models by replacing the placeholder URL on page 7 with a non-anonymous one in the future.
>
>
>
>
> **Suggestion 1: I think the calculus of how the selection probabilities are calculated from the Q-values of the candidate actions should be explicitly presented.**
>
> Thanks for the suggestion. We have now made the calculation of the action probability $P_{\mathbf{w}}$ explicit in Eqn.(5).
>
>
>
> **Suggestion 2: The temperature parameter α probably has central influence on the functioning of the method. It would be interesting to see this influence through experiments.**
>
> We have added experiments showing the impacts of alpha on several tasks in Appendix A.7. The impacts of alpha vary across tasks. Locomotion tasks seem to be more sensitive than antmaze tasks, potentially due to the larger variance of the scale of Q values in locomotion tasks. It could be possible to use an auto-tuned temperature by tracking a target entropy for $P_{\mathbf{w}}$ to make it less sensitive to the scale of Q , similar to the auto-tuning mechanism in SAC.
>
>
>
> **Suggestion 3: Also the entropy of the two policies may have a significant influence. It would be interesting to see this influence through experiments.**
>
> Entropy of prior policy $\pi_{\beta}$ is not directly controlled. And the entropy of $\pi_{\theta}$ can be controlled when SAC is used for online learning. We therefore investigated the impacts of policy entropy for PEX-SAC.
>
> We now have added experiments on this in Appendix A.7 ("Impacts of Policy Entropy"). It can be observed that impacts of policy entropy are different across tasks. Smaller or larger target entropy values could decrease the performance on some tasks, while for some tasks, better performance can be obtained by tuning the policy entropy.
>
>
>
> **Suggestion 4:**
>
> **"Some sentences could be worded more concisely...I stumbled over ...Why do you write "e.g." in italics while "et al." is not italic?**
>
>
> Thanks for these comments that further improve the presentation of the paper. We have changed all these issues as suggested in the revised paper.
>
>
>
> **Suggestion 5:**
>
> **„to a a number“** **„onehot“ -> „one-hot“** **„the policy set set as“** **„q-learning“ -> „Q-learning“** **„Should i run“ -> „Should I run“**
>
>
> Thanks for your efforts in carefully reading of our paper and pointing out a number of places that should be improved. We have changed these according to your suggestions. We have also carefully went through the reference list and fixed a few more issues along the direction pointed out by you (e.g. ImageNet, Go, t-SNE etc.). Thanks for your help in further improving the presentation of the paper.

---

> > ### Comment · Reviewer_NB3z · 2022-11-19
> > **Boltzmann**
> >
> > It is good that you have introduced Equation 5 - now it is clear that a Boltzmann distribution is used for sampling. In Equation 5 the $i$ in the sum should be replaced by another letter, e.g. $j$, because the $i$ is already used outside the sum.

---

> > > ### Author Response · Authors · 2022-11-19
> > > **Re: Boltzmann**
> > >
> > > It is great to know that we have made some previously confusing points more clear.
> > >
> > > "replace $i$ in the sum with $j$": this is a good point. We have now fixed this issue. Thanks!

---

> ### Author Response · Authors · 2022-12-02
> **Thank you for your positive feedback and for increasing the score**
>
> It is great to know that we have addressed your concerns and thank you for increasing the score.
>
> Thank you again for your efforts and constructive comments that have helped to further improve the quality of the paper.

---

### Official Review · Reviewer_EsUG · 2022-10-20

**Confidence:** 3
**Correctness:** 3
**Technical Novelty And Significance:** 2
**Empirical Novelty And Significance:** 2
**Recommendation:** 5

**Clarity, Quality, Novelty And Reproducibility:**

The overall clarity of the paper seems good, except some notation could be more clear in some specific context (such as the definition of $\textrm{Q}$).

Regarding reproducibility, the link to the code in the paper does not seem to work for me.

**Strength And Weaknesses:**

# Strength
* The idea of policy expansion, i.e., instead of using the offline policy as a warm start, one can use the offline policy as a checkpoint and combine it with the online policy (which could warm start from the offline training) and still query the checkpoint online, seems a general framework. Also as the paper already points out, the PEX framework could work agnostically to the particular online and offline RL methods.
* The experiments study is on the general D4RL benchmark, and the paper shows that PEX is competitive in most environments and outperforms some recent baselines such as JSRL and BT.
* The ablation study gives us more insights into some specific design choices such as the benefit of transferring the offline Q-function and why we should freeze the offline policy.

# Weakness
* The work seems to overlook a few more previous works in this online RL with offline dataset setup, such as [1,2,3,4].
* [5] is also very relevant to this paper's setting, and seems even more relevant than the current baselines. I wonder why [5] is not considered as in the experiment section. Also, I believe the paper should contain more explanation on why the current baselines are significant enough (I might be wrong but the baselines seem both from unpublished papers?)
* From the technical side, there are a few points that confuse me:
 1. when does $P_W$ make sense? First there is minor ambiguity in the algorithm: when it says that sampling $a \sim \tilde \pi$ according to (6), in the definition of $P_w$, $\textrm{Q}$ seems arbitrary although we can assume in the context of the algorithm it may refer to $Q_{\phi}$. However, $P_W$ is only a valid weight if $Q_{\phi}$ is accurate, otherwise, it seems just some random heuristic to select which policy to execute. But if $Q_{\phi}$ is accurate, then we should be done because an accurate $Q_{\phi}$ should implies a near optimal policy and thus we don't need to rollout and collect data anymore. Adding on this, there is no explanation of Fig.9, which records the usage of the two policies: it seems like the coefficient stays almost the same during the entire training, does this mean the $Q_{\phi}$ is already accurate at the beginning so minor updates of $Q_\phi \implies$ no changes in the amount of policy switching?

2. Another issue is why we should ever rollout the offline policy $\pi_\beta$. The ablation shows that it is beneficial to include the offline dataset into the online replay buffer, which makes perfect sense. However, the action take by $\pi_\beta$ is just a reflection of the offline data distribution, because the pessimism/penalty applied in the offline RL algorithms. Thus why should we rollout with $\pi_\beta$ because it does not seem to help with exploration/increasing data diversity?

* Finally, the stochastic policy constructed with both the offline policy/expert policy and the online policy seems relevant to the DAgger algorithm [6], with the different being that DAgger is using imitation learning loss while PEX is using RL loss. Nevertheless, it seems fair to include a discussion about such stochastic policy approaches.

[1] Todd Hester, Matej Vecerik, Olivier Pietquin, Marc Lanctot, Tom Schaul, Bilal Piot, Dan Horgan, John Quan, Andrew Sendonaris, Ian Osband, John Agapiou, Joel Z. Leibo, and Audrunas Gruslys. Deep Q-learning from demonstrations.

[2] Stephane Ross and J Andrew Bagnell. Agnostic system identification for model-based reinforcement learning.

[3] Ashvin Nair, Bob McGrew, Marcin Andrychowicz, Wojciech Zaremba, and Pieter Abbeel. Overcoming exploration in reinforcement learning with demonstrations.

[4] Mel Vecerik, Todd Hester, Jonathan Scholz, Fumin Wang, Olivier Pietquin, Bilal Piot, Nicolas Heess, Thomas Rotho ̈rl, Thomas Lampe, and Martin Riedmiller. Leveraging demonstrations for deep reinforcement learning on robotics problems with sparse rewards.

[5] Ashvin Nair, Murtaza Dalal, Abhishek Gupta, and Sergey Levine. Accelerating online reinforcement learning with offline datasets.

[6] Ross, Stéphane, Geoffrey Gordon, and Drew Bagnell. A reduction of imitation learning and structured prediction to no-regret online learning.

**Summary Of The Paper:**

The paper considers the offline-to-online RL setting and proposes policy expansion (PEX) by constructing the stochastic online policy: a mixture of the offline policy (trained using the offline data) and the online policy (trained with the online rollouts). In the experiments, the paper shows that PEX has some performance gain over some very recent methods that consider a similar setting.

**Summary Of The Review:**

Overall, there are several flaws in the current version of the paper: 1. insufficient literature review; 2. baselines not strong enough (at least no argument to show that the baselines are strong enough); 3. some technical part includes heuristic that may seem rather random. Thus I think the current version of the paper does not deliver enough contribution to the community and thus would recommend a reject at the current stage.

===========================
raised my score after the rebuttal

---

> ### Author Response · Authors · 2022-11-19
> **Response to Reviewer EsUG (Part 3/3)**
>
> **Q5: Another issue is why we should ever rollout the offline policy $\pi_{\beta}$. The ablation shows that it is beneficial to include the offline dataset into the online replay buffer, which makes perfect sense. However, the action take by $\pi_{\beta}$ is just a reflection of the offline data distribution, because the pessimism/penalty applied in the offline RL algorithms. Thus why should we rollout with $\pi_{\beta}$ because it does not seem to help with exploration/increasing data diversity?**
>
> $\pi_{\beta}$ contributes to exploration via the compositional form of the policy:
>
> -   rollout purely with $\pi_{\beta}$: if we understand it correctly, your question lies in this category, and we agree with you that in this case, rollout with $\pi_{\beta}$ may not have a great advantage in most cases. One point to note is that although the action taken by $\pi_{\beta}$ would be largely a reflection of the offline data distribution, it could also encounter novel states that are not covered by the offline dataset. Those states could potentially be useful for learning since they are not covered by the offline dataset (intuitively similar to the iterative data collection scheme in DAgger as in your next question).
>
>
> -   rollout with the policy composed from $\pi_{\beta}$ and $\pi_{\theta}$: the trajectory composition from $\pi_{\beta}$ and $\pi_{\theta}$ can be better than either one. This composition is decided by the PEX policy.
> In more detail, assume from an initial state $s_0$, rollout with $\pi_{\beta}$ has taken us to state $s$, after which we switch to unroll with $\pi_{\theta}$. Regardless of whether $s$ is well covered by the offline dataset, the states reached by $\pi_{\theta}$ after switching are less likely to be reached by unroll with a purely random policy starting from $s_0$. This is because reaching $s$ starting from $s_0$ itself is less likely with a random policy. Even in the case when $s$ is already covered by the offline dataset, this is still useful since unroll from $s$ with $\pi_{\theta}$ also has a good chance of leading to novel states that are not covered by the offline dataset. From this perspective, unroll with $\pi_{\beta}$ can be intuitively interpreted as a way to set the simulator to a state $s$ (which could be already covered in offline data), and then explore from there. The compositional form of policy leads to this type of jump-start form of exploration, where $\pi_{\beta}$ is adaptively involved.
>
>
> -   This is just a simplified example for explanation purpose and the actual way of composition is adaptively determined using $Q_{\phi}$.
>
>
>
>
> **Q6: The stochastic policy constructed with both the offline policy/expert policy and the online policy seems relevant to the DAgger algorithm [6], with the different being that DAgger is using imitation learning loss while PEX is using RL loss. Nevertheless, it seems fair to include a discussion about such stochastic policy approaches.**
>
> Thanks for pointing out this interesting connection.
>
> -   DAgger indeed also uses a compositional form of policy, although with a uniform weight across states and under a different context of imitation learning as you noted. In our case the compositional weight is state adaptive and the compositional policy is used for a very different purpose of bridging offline to online reinforcement learning. We have added discussion on this interesting connection after Eqn.(6) in the updated paper.
>
>
> -   In addition, we feel this connection is also related to the discussion in Q5, offering further hints on rollout with $\pi_{\beta}$ in a compositional form is one effective way to leverage it for data collection/exploration.
>
>
>
>
>
>
> **Q7: The link to the code in the paper does not seem to work for me.**
>
> The current link is a placeholder URL. We will replace it with a non-anonymous one in the future for code release to facilitate fellow researchers in the community.
>
>
>
>
>
> **References**
>
> [1] Todd Hester, Matej Vecerik, Olivier Pietquin, Marc Lanctot, Tom Schaul, Bilal Piot, Dan Horgan, John Quan, Andrew Sendonaris, Ian Osband, John Agapiou, Joel Z. Leibo, and Audrunas Gruslys. Deep Q-learning from demonstrations.
>
> [2] Stephane Ross and J Andrew Bagnell. Agnostic system identification for model-based reinforcement learning.
>
> [3] Ashvin Nair, Bob McGrew, Marcin Andrychowicz, Wojciech Zaremba, and Pieter Abbeel. Overcoming exploration in reinforcement learning with demonstrations.
>
> [4] Mel Vecerik, Todd Hester, Jonathan Scholz, Fumin Wang, Olivier Pietquin, Bilal Piot, Nicolas Heess, Thomas Rotho ̈rl, Thomas Lampe, and Martin Riedmiller. Leveraging demonstrations for deep reinforcement learning on robotics problems with sparse rewards.
>
> [5] Nair et al., AWAC: Accelerating Online Reinforcement Learning with Offline Datasets, 2020
>
> [6] Ross, Stéphane, Geoffrey Gordon, and Drew Bagnell. A reduction of imitation learning and structured prediction to no-regret online learning.

---

> > ### Comment · Reviewer_EsUG · 2022-12-13
> > **Response**
> >
> > I appreciate the author's response. The experiments are more complete after the revision. Also the authors better placed their work in the literature after the revision. However, although the authors put a sincere effort in trying to explain their algorithm design (which I greatly appreciate), I still believe one or two algorithm design details are not very intuitive and I still have a hard time understanding why those parts would be beneficial. I have updated my score accordingly.

---

> > > ### Author Response · Authors · 2022-12-13
> > > **To Reviewer EsUG: Thank you for your encouraging feedback and for raising the score**
> > >
> > > We thank reviewer EsUG for your encouraging feedback on our rebuttal and for raising the score.
> > >
> > > From your feedback, it is great to know that we have successfully addressed your original concerns on literature and baselines. And thank you for your appreciation of our efforts on explaining the algorithm.
> > >
> > > For the last remaining concern on “one or two algorithm design details are not very intuitive”, although not very clear to us what they are at this point, we believe they are related to the original comment that some “heuristic that may seem rather random”.
> > >
> > > Here we only want to summarize briefly our efforts on addressing the original concern that some “heuristic that may seem rather random”, in case the big picture was obfuscated in the original rebuttal (apologize for the lengthy explanations in original rebuttal):
> > >
> > > 1. To address this comment, we have provided some intuitive interpretations and derivations of $P_\mathbf{w}$ in Appendix A3. This newly added derivation can serve as additional evidence showing that the $P_\mathbf{w}$ is not a random heuristic.
> > > 2. In addition, detailed and intuitive interpretations on $P_\mathbf{w}$ and $\pi_{\beta}$ are provided in the reply to Q3 and Q5.
> > >
> > > Although it appears to us that the original concern that some “heuristic that may seem rather random” has been sufficiently addressed from our perspective, it’s possible that there are some other points that we are unaware of are not well explained.
> > > We are happy to provide further explanations on any concrete points that have the potential of confusion.
> > >
> > > We thank Reviewer EsUG again for your time and valuable comments that have helped improve the quality of the paper.

---

> ### Author Response · Authors · 2022-11-19
> **Response to Reviewer EsUG (Part 2/3)**
>
>
> **Q3: when does $P_{\mathbf{w}}$ make sense ... $P_{\mathbf{w}}$ is only a valid weight if $Q_{\phi}$ is accurate, otherwise, it seems just some random** **heuristic** **to select which policy to execute. But if $Q_{\phi}$ is accurate, then we should be done because an accurate $Q_{\phi}$ should implies a near optimal policy and thus we don't need to rollout and collect data anymore.**
>
> Sorry for the confusion. By $Q$, we indeed meant $Q_{\phi}$ in Eqn.(6). Thanks for the comment and we have now made the $\phi$ parameter explicit.
>
>
> There seems to be some confusion on $P_{\mathbf{w}}$ which we want to clarify in the following:
>
>    -   $P_{\mathbf{w}}$ is not a random heuristic. We have now added some intuitive interpretations and derivation of it in Appendix A3. $P_{\mathbf{w}}$ can be viewed as a discrete policy with action dimension of two, which selects between two candidate actions. Ideally, it assigns higher probability to actions with higher values at the current state. Eqn.(6) is essentially a reflection of this intuition and can be derived as shown in Appendix A3.
>
> - $Q_{\phi}$ does not need to be accurate to start learning:
>    -  In standard RL, we start from a random policy $\pi$ and random $Q_{\phi}$ (which is far from accurate) and then learn a better $Q_{\phi}$ through TD-learning based on the transitions collected by interacting with environment using action sampled from $\pi(s)$ given the current observation $s$. In the mean time, the policy can be improved based on the improved $Q_{\phi}$.
>    -   Similar to the reasoning above, $P_{\mathbf{w}}$ (or $Q_{\phi}$) does not need to be accurate to start learning. Naturally, sampling from $P_{\mathbf{w}}$ is part of the exploration process, as in typical RL. $P_{\mathbf{w}}$ does not need to be perfect to be used in exploration and learning. A better $Q_{\phi}$ will be learned via TD-learning based on the collected transitions, which in turn improves $P_{\mathbf{w}}$ since it is constructed from $Q_{\phi}$.
>
> - The level of accuracy of $Q_{\phi}$ can be roughly interpreted as the amount of prior knowledge encoded.
>
>      -   When $Q_{\phi}$ is random, $P_{\mathbf{w}}$ will also be random, and we reduce to the *tabula rasa* case. In this case, learning can still happen, just no injected prior knowledge;
>
>      -   When $Q_{\phi}$ is perfectly accurate, it has encoded all the useful knowledge already. In this case, no further learning is required (we agree with you on this);
>
>      -   When $Q_{\phi}$ is not perfectly accurate which is the most typical case in practice, it can still provide useful guidance on action selection in some cases, and might be less informative in some other cases. Along the learning process, $Q_{\phi}$ will be more and more accurate, and the guidance will be more and more informative.
>
>  In summary, $Q_{\phi}$ does not need to be accurate to start learning, but a more accurate $Q_{\phi}$ encodes better/more prior knowledge which could potentially result in larger speedup in learning.
>
>
>
>
> **Q4: There is no explanation of Fig.9, which records the usage of the two policies: it seems like the coefficient stays almost the same during the entire training, does this mean the $Q_{\phi}$ is already accurate at the beginning so minor updates of $Q_{\phi}$⟹ no changes in the amount of policy switching?**
>
> We apologize that we didn't provide enough context information for Fig 9 in the original submission. Originally, those figures are the visualization of policy usages of BC-PEX (the policies pre-trained with BC), as corresponding to the experiments shown in Fig. 8 of the original paper. To generate those figure, we smoothed the curve and averaged across 5 runs. The fluctuations are less perceivable because of these operations.
>
> We have now regenerated the figures (now in Figure 11 of the updated paper) and also added figures for IQL-PEX (PEX pre-trained with IQL) to be more complete, with more explanations. The updated figures are generated from one training session (no averaging over multiple runs) without excessive smoothing. It can be observed that there are clear fluctuations along the processing of training, meaning the amount of policy switching is changing during training, along with the learning of $Q_{\phi}$.

---

> ### Author Response · Authors · 2022-11-19
> **Response to Reviewer EsUG (Part 1/3)**
>
> We thank the reviewer for the constructive review and insightful comments. Responses to the questions are below:
>
>
>
> **Q1: The work seems to overlook a few more previous works in this online RL with offline dataset setup, such as [1,2,3,4].**
>
> Thanks for pointing out more previous works. This comment helps to further expand the related work and reveals interesting connection with our baseline. Both help to further improve the quality of the paper.
>
> In the original submission we have mainly focused on the related work in the offline RL setting where the offline data are general (e.g. from random policy) and is not necessarily demonstrations. Now we have now included the suggested literatures on demonstration data according to you suggestion, further expanding the related work and providing an even broader context of literature.
>
> We have carefully read the suggested references and and added citations at multiple appropriate places in the updated paper (introduction, related work, experiments etc) with some discussions in related work section.
>
> Of particular, [4] is an approach to load and keep the the offline data for online training. It indeed corresponds to our "Buffer" baseline in section 6.1. Thanks for pointing out this work, which backs up our designed baseline and makes it better grounded in literature.
>
>
>
>
>
> **Q2: [5] is also very relevant to this paper's setting, and seems even more relevant than the current baselines. I wonder why [5] is not considered as in the experiment section. Also, I believe the paper should contain more explanation on why the current baselines are significant enough (I might be wrong but the baselines seem both from unpublished papers?)**
>
> AWAC [5] is indeed very relevant to our work and that's also why we have cited it at multiple places in our work (e.g. Section 2.2 "Policy Learning from Offline Dataset", and Section 5 "Offline Training with Online Fine-tuning").
>
> We want to explain that our baselines are significant, representative and comprehensive:
>
> -   significance of baselines:
>
>     -   AWAC was not used as baseline in the original submission since we have already included IQL (denoted as "Direct" in Sec. 6.1), which is a stronger baseline. As we discussed in subsection "Offline Training with Online Fine-tuning" of Section 5, the proposed approach falls into the category of "offline-to-online RL". Representative previous works in this category include AWAC and IQL. IQL is a recent published work (ICLR22) that achieves state-of-the-art performance for offline RL while also allows online fine-tuning, outperforming AWAC according to the IQL paper.
>
>
> -   representativeness and comprehensiveness of baselines:
>
>     -   Beyond this, we also included a number of other baselines because of their connections to the proposed approach for more comprehensive evaluations. Each of them highlights different aspects, making the baselines representative. For example, "BT" and "JSRL" are baselines on different ways of utilizing offline policy for exploration. "Buffer" (a baseline that actually corresponds to the approach in [4] as revealed by your first comment) represents a baseline on using offline data directly without pre-training.
>
>
> We agree with you that it is useful for include AWAC's results for reference and we have now added AWAC as another baseline according to your suggestion. The aggregated AWAC results are added to Fig 3 and Fig 4. The individual curves of AWAC are added to Fig 2. This makes our set of baselines even more complete and comprehensive.

---

### Official Review · Reviewer_2rLA · 2022-10-23

**Confidence:** 4
**Correctness:** 3
**Technical Novelty And Significance:** 3
**Empirical Novelty And Significance:** 3
**Recommendation:** 6

**Clarity, Quality, Novelty And Reproducibility:**

The paper is overall easy to follow but makes many less relevant claims such as expanding policy set. The originality of the work is good. The reproducibility is unclear as the code is not provided.

**Strength And Weaknesses:**

Strength:

1. The idea of using a policy set to reduce the distributional shift issue and conservatism when switching from offline to online training is neat.
2. The empirical results are pretty impressive. the ablation studies are also quite thorough, clearly suggesting the benefits of different components of the algorithm.

Weaknesses:

1. I think the authors should compare the method to more offline-to-online works such as Lee et al., 2021, Yang et al,. 2021, Lu et al., 2022, Zheng et al., 2022 and etc. I think it is important to see if the method can do better than approaches that are not like IQL.
2. The story of using a policy set seems a bit disconnected to the main point of the paper. The main idea of the paper is essentially to train a different online policy and maintain the offline policy for exploration whereas the policy set idea is more than this. I think the authors should make the writing more clear and to the point rather than making too many less relevant statements.
3. The authors don't provide much theoretical insight of the method.
4. The error bars of the experiments are quite high. I wonder if the authors should run more random seeds / do hypothesis testing to make sure the results are statistically significant.

**Summary Of The Paper:**

This paper presents a new offline-to-online finetuning approach that proposes to maintain both the policy learned offline and the new online policy during online-finetuning. The authors first train the offline policy and Q-function, select the policy from a categorical distribution with the learned Q-values during online data collection, and optimize the online policy using data sampled from both the offline dataset and the newly collected online samples. The offline policy is kept frozen during the finetuning step. The authors show that doing so can alleviate the distributional shift and over-conservatism issues when switching from offline training to online training with a single policy. Empirical results suggest that the method can outperform prior method IQL and its variants on the D4RL benchmark.

**Summary Of The Review:**

Based on my comments above, I vote for a weak accept of the paper.

---

> ### Author Response · Authors · 2022-11-19
> **Response to Reviewer 2rLA**
>
> We would like to thank the reviewer for the positive review and for the efforts in reviewing our work. Your comments have helped to further improve the quality of the paper.
>
> Responses to the questions are below:
>
> **Q1: I think the authors should compare the method to more offline-to-online works such as Lee et al., 2021... I think it is important to see if the method can do better than approaches that are not like IQL.**
>
> Thanks for the suggestion. We have now added comparison with the offline-to-online work by Lee et al., 2021 [1] suggested by you (referred to as "Off2On" in our updated paper). We have also added citations to all the suggested references.
>
> The individual curves of Off2On are added to Fig 9 since it is SAC-based for online learning. The aggregated results are added to Fig 3 and Fig 4. Off2On is a strong baseline that achieves good overall performance, and PEX still outperforms it in terms of the overall performance.
>
> In addition to Off2On, we have also added another offline-to-online work AWAC [2], which is another representative work for offline-to-online RL.
>
>
>
> **Q2: The story of using a policy set seems a bit disconnected to the main point of the paper. The main idea of the paper is essentially to train a different online policy and maintain the offline policy for exploration whereas the policy set idea is more than this.**
>
> We use policy set as a succinct notion to represent the collection of both policies, from which the final policy is adaptively composed as shown in Eqn.(6). It is important to note that the offline policy is not only used for exploration, but is also part of the final policy and the policy set representation is convenient to reflect this fact. Therefore we keep the notion of policy set. Please let us know if you have any further suggestion on this.
>
> We do agree with you that the policy set idea could potentially be applied to other cases as well (e.g. more than two policies etc.). And it would indeed be interesting to explore its application in other cases beyond the domain of this work. We have now added this point as one of our future work in the updated paper inspired by your comment.
>
>
>
> **Q3: The authors don't provide much theoretical insight of the method.**
>
> We have now provided some intuitive interpretations and derivation of the proposed method in Appendix A3.
> $P_{\mathbf{w}}$ can be interpreted as a higher level discrete policy for selecting candidate actions. The fact that $P_{\mathbf{w}}$ is constructed from $Q$ makes the action selecting more informed by leveraging the knowledge learned during the offline pre-training of $Q$. $\pi_{\beta}$ functions as one source of action proposals, providing useful samples that can generate actions with reasonably high Q values compared with random actions, which could help with learning.
>
>
>
>
>
> **Q4: The error bars of the experiments are quite high.**
>
> Different tasks and methods indeed have different level of variations. To facilitate comparison, we have not only repeated the experiments with a reasonable number seeds (5 in our case), but also expand the task set by including 15 different tasks. The aggregated curves over tasks is actually with more clear and comprehensive in comparing different algorithms than the curves for each individual task. And that's why we have provided the aggregated curves for method comparison and ablations.
>
>
>
>
> **References**
>
> [1] Lee et al., Offline-to-Online Reinforcement Learning via Balanced Replay and Pessimistic Q-Ensemble, 2021
>
> [2] Nair et al., AWAC: Accelerating Online Reinforcement Learning with Offline Datasets, 2020

---

### Official Review · Reviewer_U35N · 2022-10-25

**Confidence:** 3
**Correctness:** 4
**Technical Novelty And Significance:** 2
**Empirical Novelty And Significance:** 3
**Recommendation:** 6

**Clarity, Quality, Novelty And Reproducibility:**

Overall, the paper is well-written and easy to follow. The idea of policy expansion and its combination with adaptive policy composition seems novel, but it is a rather heuristically designed method, rather than a principled one.

**Strength And Weaknesses:**

[Strengths]
1. This work provides a simple yet effective scheme for offline-to-online RL. The proposed method is easy to implement.
2. In the experiments, PEX significantly outperforms baseline algorithms.

[Weaknesses]
1. In the experiments, comparisons with existing methods for offline-to-online RL are missing, e.g. [1,2].
2. To claim that PEX is a general framework, rather than an IQL-specific algorithm, it would be great to demonstrate the effectiveness of diverse combinations of PEX and existing offline RL algorithm backbones. For example, it would be great to see the performance of CQL+PEX or FisherBRC+PEX and whether PEX can consistently outperforms the baselines even for those different backbones.
3. The proposed method is designed heuristically, without theoretical justification.

[Questions]
1. In the PEX framework, the offline training phase is to learn a Q function and a policy, where the policy should be frozen while the value should be fine-tuned (i.e. transfer Q). I am wondering why value-finetuning is beneficial but policy-finetuning is detrimental to offline-to-online performance. It would be great to provide a more detailed explanation about this.
2. In Figure 2, why is the performance of PEX in halfcheetah-random better than in halfcheetah-medium-replay even though medium-replay dataset would contain more high-reward experiences?
3. It would be great to see the result using Q-transfer only without the policy expansion, i.e. initialize the Q-function with the pretrained one, and run an online RL algorithm using the Q initialization.
4. How sensitive is the algorithm with respect to the hyperparameter alpha?


[1] Nair et al., AWAC: Accelerating Online Reinforcement Learning with Offline Datasets, 2020

[2] Lee et al., Offline-to-Online Reinforcement Learning via Balanced Replay and Pessimistic Q-Ensemble, 2021



**Summary Of The Paper:**

This paper presents a scheme of policy expansion (PEX) to bridge offline and online RL. Instead of directly fine-tuning the policy learned by offline dataset during online interaction, PEX freezes the learned policy and expands the policy set with a newly added policy. This new policy is optimized during the online learning phase. Then, a single policy is derived from the policy set, where action candidates are sampled by each policy and then the final action is selected by a resampling distribution. In the D4RL benchmark, PEX overall outperforms all of the baseline algorithms.


**Summary Of The Review:**

The idea of policy expansion is interesting, but it lacks theoretical justification. For the experiments, it would be great to make a comparison with more baselines (see Weaknesses part) and to see the generality of the proposed PEX framework by conducting experiments with different offline RL backbones.

== post-rebutal
Thank you very much for your detailed responses. The added experiments are also greatly appreciated. I raised my score accordingly.

---

> ### Author Response · Authors · 2022-11-19
> **Response to Reviewer U35N (Part 2/2)**
>
> **Q2: "why is the performance of PEX in halfcheetah-random better than in halfcheetah-medium-replay even though medium-replay dataset would contain more high-reward experiences?"**
>
> Thanks for your question.
> -   It is interesting to note that this observation is consistent with those of other researchers. For example, Lee et al. reported in their paper [2] (page 18 "Dataset composition and performance" and Fig 13): "... ran our method on halfcheetah tasks for 1 million steps ... our method performed best on random dataset" while the performance on medium and medium-replay are lower.
>
>
> -   While it is hard to pinpoint one single reason for this counterintuitive observation, we conjecture that it is mainly due to the task's sensitivity to initial data coverage/diversity. By that, we mean the coverage/diversity of the initial data the policy is trained on. halfcheetah task is known to be sensitive to the initial data diversity. In fact, from our own empirical experience on other projects, we observed that a larger number of random steps for initial collection could improve performance and stability for running standard online RL (e.g. for SAC) on halfcheetah task, suggesting its sensitivity to data coverage.
>
>
> -   Back to our offline-to-online scenario, the offline dataset could have a similar impact as the initially collected data does on online RL learning. The data in medium-replay from D4RL is the replay buffer when an RL agent is trained to reach a medium level. Although there is a certain level of diversity due to the evolution of policy along training, its diversity might be not as large as the random one.
>
>
> -   Furthermore, the benefits of the high-reward experiences in the medium-replay dataset may not be as prominent as in sparse reward case, since halfcheetah task is a dense reward task, and the agent can quickly collect experiences with good reward when interacting with the environment during the online stage.
>
>
>
>
> **Q3: "It would be great to see the result using Q-transfer only without the policy expansion, i.e. initialize the Q-function with the pretrained one, and run an online RL algorithm using the Q initialization."**
>
> Thank you for the great suggestion. We have now added the results according to your suggestion in Section 6.3 and Figure 5, denoted as "Policy Transfer".
>
> Basically, by disabling policy transfer in PEX, we arrive at your suggested ablation "using Q-transfer only without the policy expansion". The results show that this ablation baseline performs worse than the full PEX method, implying the importance of policy expansion.
>
> This good suggestion makes our current ablations more symmetric and comprehensive by covering both Critic Transferring and Policy Transferring.
>
>
>
> **Q4: "How sensitive is the algorithm with respect to the** **hyperparameter** **alpha?"**
>
> We have added experiments showing the impacts of alpha on several tasks in Appendix A.7. The impacts of alpha vary across tasks. Locomotion tasks seem to be more sensitive than antmaze tasks, potentially due to the larger variance of the scale of Q values in locomotion task. It is possible to use auto-tuned temperature by tracking a target entropy for $P_{\mathbf{w}}$ to make it less sensitive to the scale of Q, similar to the auto-tuning in SAC.
>
>
>
>
> **References**
>
> [1] Nair et al., AWAC: Accelerating Online Reinforcement Learning with Offline Datasets, 2020
>
> [2] Lee et al., Offline-to-Online Reinforcement Learning via Balanced Replay and Pessimistic Q-Ensemble, 2021
>
> [3] Lyu et al., Mildly Conservative Q-Learning for Offline Reinforcement Learning 2022

---

> ### Author Response · Authors · 2022-11-19
> **Response to Reviewer U35N (Part 1/2)**
>
> We thank the reviewer for the review and constructive comments that help improved the quality of the paper from several aspects.
>
> Responses to the questions are below:
>
>
>
> **Weakness 1: comparisons with existing methods for offline-to-online RL are missing,** **e.g.** **[1,2]**
>
> Thanks for the suggestion. In the original submission, we have mainly compares with IQL (denoted as "Direct" in Sec. 6.1) and variants based on it. IQL is a recent work that achieves good performance for offline RL while also allows online fine-tuning, outperforming AWAC according to the IQL paper.
>
> We agree with you that it is also useful to include other offline-to-online RL methods to offer a more comprehensive comparison. We have now added comparison with AWAC [1] and Off2On [2] according to your suggestion. The aggregated results are added to Fig 3 and Fig 4. The individual curves of AWAC [1] are added to Fig 2 since it uses a similar type of policy update as IQL. The individual curves of Off2On [2] are added to Fig 9 since it is SAC-based for online learning.
> Off2On is a strong baseline that achieves good overall performance, and PEX still outperforms it in terms of the overall performance (Fig 3).
>
>
>
> **Weakness 2: “it would be great to demonstrate the effectiveness of diverse combinations of PEX and existing offline RL algorithm backbones"**
>
> We want to clarify that we meant the framework is *compatible* to be used together with different value-based offline and online RL algorithms. However, there is no guarantee that any such compatible combination could be equally effective.
> We have updated the paper to make it more clear.
>
> Actually, the final performance could be impacted by the selection of offline RL methods. There are already some evidences on this in literature. E.g., in [1], the authors reported that the excessively conservative offline training could lead to slow fine-tuning. It's possible that plugging in this type of offline RL methods could have a similar impacts on PEX. On the other hand, it would be interesting to see how much we can gain by upgrading it on offline RL together with different online methods (e.g. "mildly conservative" methods [3]). Since our limited computational resources have been allocated to run the suggested new baselines (AWAC, Off2On) and various ablations (policy transfer, temperature etc), we leave this largely orthogonal theme of large scale evaluation different combinations and searching for the best as a future work.
>
>
> **Weakness 3: “The proposed method is designed heuristically, without theoretical justification"**
>
>
> $P_{\mathbf{w}}$ can be interpreted as a higher level discrete policy for selecting candidate actions. The fact that $P_{\mathbf{w}}$ is constructed from $Q$ makes the action selecting more informed by leveraging the knowledge learned during the offline pre-training of $Q$. $\pi_{\beta}$ functions as one source of action proposals, providing useful samples that can generate actions with reasonably high $Q$ values compared with random actions, which could help with learning.
>
> We have now provided theoretical justification of the proposed method in Appendix A.3.
>
>
>
>
> **Q1: “In the PEX framework ... the policy should be frozen while the value should be fine-tuned (i.e. transfer Q). I am wondering why value-finetuning is beneficial but policy-finetuning is detrimental to offline-to-online performance."**
>
> - After switching from offline to online learning, _both_ the value function and the policy should be adjusted according to new experiences for further improvement. However, as also noted in previous work [1, 2], the way to adjust the policy is critical and requires care. By directly fine-tuning policy parameters (referred to as "Direct"), initial noisy gradient updates (due to the distribution shift between offline and online data etc.) to the policy could negatively impact or even completely destroy in extreme cases the previously learned good behavior, largely discarding potential benefits of the offline pre-training on exploration and sample efficiency etc. (e.g. as also mentioned in[2] "updates in such unseen regime ... destroys the good initial policy obtained via offline RL").
>
> -   To address this issue, different schemes have been developed in the community. For example, [2] used a balanced replaying scheme to modulate the trade-off between using online and offline samples to make the samples more on policy while maintaining a diversity.
>
>
> -   In PEX, we preserve a snapshot of the offline policy. If the offline policy is not freezed, it will roughly reduce to a specific instance of the "Direct" approach discussed above, with a specially structured policy network. Therefore, it could also compromise its performance because of the same reasons faced by other "Direct" approaches. Therefore, we propose to avoid compromising the previously learned behavior by freezing it and improve over it by learning with the newly added policy $\pi_{\theta}$ in PEX.

---

> ### Author Response · Authors · 2022-12-08
> **To Reviewer U35N: Thank you for your positive feedback and for raising the score**
>
> We thank Reviewer U35N for your positive feedback on the rebuttal and thank you for raising the score. We greatly appreciate your efforts in reviewing the paper and your valuable comments that have helped to further improve the quality of the paper.

---

### Author Response · Authors · 2022-11-19
**Thanks for the reviews and summary of key paper changes**

We thank all the reviewers (**R1**-U35N, **R2**-2rLA, **R3**-EsUG and **R4**-NB3z) for their efforts in reviewing the paper and their constructive comments that help to further improve the quality of the paper.

The reviewers appreciated the:

1.  novelty of the method (**R1**-U35N, **R2**-2rLA, **R4**-NB3z)


2.  interestingness of the idea (**R1**-U35N, **R2**-2rLA, **R4**-NB3z)


3.  good results (**R1**-U35N, **R2**-2rLA, **R3**-EsUG, **R4**-NB3z)


4.  clear presentation and is easy to follow (**R1**-U35N, **R2**-2rLA, **R3**-EsUG, **R4**-NB3z).


We have further improved the paper according to the comments from all reviewers. We summarize in the following the key changes made to the paper. We added

1.  two more baselines: AWAC [Nair20], and Off2On [Lee21]
2.  more experiments on the impacts of some hyper-parameters (temperature and entropy)
3.  additional ablation results (policy transfer)
4.  citations to suggested references with some discussions
5.  interpretation and derivation of the proposed approach (Appendix A.3 "theoretical justification of PEX")


The major content changes in the revised paper are highlighted in blue for easier review.

---

### Decision · Program_Chairs · 2023-01-20

**Decision:**

Accept: poster

**Justification For Why Not Higher Score:**

The weaknesses listed above:
- The algorithm designs are not very intuitive here and there (1) why freeze the offline policy rather than fine-tune it? (2) why use online Q-function to evaluate the quality of offline action? i.e. use offline Q-function to evaluate offline actions and online Q-function for online actions?
- Readability: (1) the policy set contains just two policies, so why just call it a "policy pair" or something similar? (2) A.4 contains only one study but the section title and main text says it contains result"s".

**Justification For Why Not Lower Score:**

- The remarkably simple idea that works very well against baselines

**Metareview: Summary, Strengths And Weaknesses:**

This paper addresses the problem of jump-starting reinforcement learning with the dataset collected a-priori, known as offline-to-online reinforcement learning. The main idea is to model the policy as a mixture model with two components, where the first component is the policy pre-trained by an offline RL algorithm and the second component is the target policy being learned online. The pre-trained policy is kept fixed, while the target policy is updated by online interaction data. The mixture weight is determined by the softmax distribution of Q-values. This remarkably simple idea is shown to perform very well on standard benchmark tasks.

However, there are certainly weaknesses in a number of areas:
- The algorithm designs are not very intuitive here and there (1) why freeze the offline policy rather than fine-tune it? (2) why use online Q-function to evaluate the quality of offline action? i.e. use offline Q-function to evaluate offline actions and online Q-function for online actions?
- Readability: (1) the policy set contains just two policies, so why just call it a "policy pair" or something similar? I guess the authors initially designed the method to progressively increase the set over training epochs? In this sense, the terminology "policy expansion" sounds too much, and perhaps "dual policy framework" would sound better?  (2) A.4 contains only one study but the section title and main text says it contains result"s".





**Note From Pc:**

if the above contains the word "oral" or "spotlight" please see: "oral" presentation means -> notable-top-5% and "spotlight" means -> notable-top-25%. As stated in our emails, we are disassociating presentation type from AC recommendations

**Summary Of Ac-Reviewer Meeting:**

Overall, reviewers appreciated the effectiveness yet the simplicity of the the approach proposed in the paper, which was the main point behind recommending acceptance of the paper.

Although the paper improved quite a lot through revisions, reviewers pointed out there was still room for improvement, listed above in the meta-review.